# Wide field light-sheet microscopy with lens-axicon controlled two-photon Bessel beam illumination

Sota Takanezawa[1,3], Takashi Saitou [1,2,3✉] & Takeshi Imamura[1,2]

Two-photon excitation can lower phototoxicity and improve penetration depth, but its narrow excitation range restricts its applications in light-sheet microscopy. Here, we propose simple illumination optics, a lens-axicon triplet composed of an axicon and two convex lenses, to generate longer extent Bessel beams. This unit can stretch the beam full width at half maximum of 600–1000 μm with less than a 4-μm waist when using a 10× illumination lens. A two-photon excitation digital scanned light-sheet microscope possessing this range of field of view and ~2–3-μm axial resolution is constructed and used to analyze the cellular dynamics over the whole body of medaka fish. We demonstrate long-term time-lapse observations over several days and high-speed recording with ~3 mm$^3$ volume per 4 s of the embryos. Our system is minimal and suppresses laser power loss, which can broaden applications of two-photon excitation in light-sheet microscopy.

[1] Department of Molecular Medicine for Pathogenesis, Graduate School of Medicine, Ehime University, Matsuyama, Japan. [2] Translational Research Center, Ehime University Hospital, Toon, Japan. [3] These authors contributed equally: Sota Takanezawa, Takashi Saitou. ✉email: t-saitou@m.ehime-u.ac.jp

Modern advanced fluorescence microscopies such as confocal microscopy and light-sheet microscopy play invaluable roles in elucidating single-cell behaviors of multicellular organisms[1,2]. In combination with fluorescent labeling techniques, three-dimensional (3D) time-lapse imaging of living tissues over whole animals can clarify how cellular circuits organize and function over time, helping to elucidate a system-level understanding of cell population activities[3]. These types of whole organism level analysis require a high resolution such as cellular level and large field of view (FOV) which covers the whole tissue or organs imaging.

Cellular level imaging has been realized due to optical sectioning techniques. Laser scanning confocal microscopy has achieved optical sectioning by scanning focused laser beams and performing point-by-point detection of the emission fluorescence with a pinhole to exclude out-of-focus signals[4]. Microscope systems generally have tradeoffs between the spatial resolution, image acquisition speed, and photodamage. Scanning-type measurements are time-consuming for 3D image acquisition due to their non-imaging detection of fluorescence. Furthermore, this microscopy uses the epi-illumination method, which exposes the entire sample to radiation. Consequently, it causes a phototoxic effect to living tissues, limiting long-term live imaging applications. Light-sheet microscopy is a different cutting-edge technology, which offers a higher acquisition speed with a lower phototoxicity. Hence, it is a vital tool for time-lapse observations over long periods of time. It employs a different optical sectioning technique with separate illumination and detection pathways. The sample is illuminated with a plane of light, while imaging occurs through a sensitive camera oriented orthogonally to the light sheet[5–7]. Due to this thin selective plane illumination, the total fluorophore excitation is greatly reduced compared to the confocal microscopy, resulting in low photobleaching and phototoxic effects. Since the shape of the illumination sheet and the detection machineries determine the spatial resolution and the FOV in this microscopy, the optics need to be carefully balanced. There is a trade-off between FOV and the axial resolution due to the orthogonal geometry of the illumination and detection pathways. A light-sheet created by a Gaussian beam with a thin waist offers high resolution along the axial direction, but decreases the propagation length along the illumination axis, which prevents large FOV imaging. Conversely, a Gaussian beam with a thicker waist provides a longer propagation length, but worsens the resolution. In addition, when the illumination light propagates through biological samples, scattering and absorption of the light-sheet create irregularly patterned dark and bright stripes due to the inhomogeneous distribution of objects. This decreases the penetration deep inside tissues and degrades the image contrast. This makes in vivo imaging over the entire sample challenging.

To overcome these image contrast limitations, several approaches have been proposed, including the use of two-sided excitation[8,9], multi-view observation[10,11], and structured illumination[12]. Two light sheets illuminated from opposite sides compensate for the decreased penetration depth deep inside tissues, enabling the observation of larger samples. Multi-view images combined with image fusion techniques from different angle images also can reduce the image anisotropy and enhance the image quality. Moreover, structured illumination can remove out-of-focus excitation signals and increase the spatial resolution. However, these methods require additional light exposure to samples and image acquisition procedures at the expense of phototoxicity, photobleaching, and image acquisition speed for constructing final images.

Another approach to increase the image quality is to use non-diffracting beams[13–16] such as a Bessel beam. A Bessel beam is created through an annular pattern at the back focal plane of the illumination objective (IO) lens. Its central peak has a needle shape in the direction of light propagation, expanding the focus depth of excitation and improving the excitation unevenness. However, some of the energy is distributed in the side lobes of the beam, which appear as surrounding rings. The distribution profile of the central and side lobes is related to the propagation length in that suppressed energy in the side lobes generates relatively short propagation beams, and vice versa[17,18]. This is a trade-off in Bessel beams. An advantage of the Bessel beam is a self-reconstruction property, which inhibits shadows behind objects and can propagate deeper into biological samples[18–21]. However, the side lobes of the Bessel beams produce extra photobleaching and out-of-focus fluorescence, which causes blurred background signals. Exclusion of these signals can be overcome by a combination of a Bessel beam and two-photon excitation[17,22–24] because the quadratic dependence of the fluorescence excitation suppresses the side lobe signals compared with the central lobe signals. Two-photon excitation technology using near-infrared lasers is beneficial for the penetration depth to a living sample and reducing photodamage[25,26]. The reported advantages of using a two-photon excitation in light-sheet microscopy are improved penetration depth and background rejection[22]. Therefore, the use of Bessel-type near-infrared lasers for two-photon excitation extends the propagation length while maintaining a thin beam waist and improves both the penetration depth and scattering of the light sheet due to the biological tissue structure. Its utility has been demonstrated for observations of subcellular structures[17], and living multicellular organisms of *Caenorhabditis Elegans*[21], zebrafish[27], and tumors with microenvironments[18,19]. Previous reports typically employed short beams with high numerical aperture (NA) IO lenses, in which the reported maximum FOV is 600 μm with a beam waist of 2 μm[27]. The use of lower NA objective lenses makes it possible to obtain a larger FOV, but the efficiency of the nonlinear fluorescence excitation should be reduced. Therefore, techniques that deliver the sufficient laser energy to the IO lens to induce two-photon excitation with a lower NA objectives over various beam lengths are useful for imaging large samples and fast events with a high spatial resolution.

There has been a growing interest in observing the whole body of animals to study biological and disease processes which occurred across the entire body. Medaka is a suitable animal for such an analysis. Medaka, *Oryzias latipes*, is a small freshwater teleost fish. This fish as well as zebrafish have been used as model organisms in fields of developmental, molecular, and evolutionary biology, etc.[28,29]. These species possess several common characteristics such as small size, egg laying, high fecundity, short generation times (2–3 months), and available genome resources. On the other hand, the differences between these species have been characterized. Medaka tolerates a wide variety of salinities and temperatures. Embryos hatch after 7–8 days, while zebrafish embryos hatch 3 days post fertilization. Furthermore, due to high tolerance to inbreeding, lots of isogenic inbred strains have been established in medaka, which serve as unique genetic resources. These features distinguish medaka from zebrafish as a model organism[30]. Therefore, technologies that allow us to comprehensively analyze the organism phonotype at the cellular-to-organ level would be of special importance in genetic research. These fishes also have several advantages in terms of microscopic observations: because (1) they are egg laying, embryonic development occurs externally, (2) embryos are transparent, and (3) their size ~1 mm. However, medaka embryos (~1.2 to 1.3-mm) are larger than zebrafish embryos (~0.8 mm), fly, and *C. Elegans*. The characteristics of size and slow developmental time make

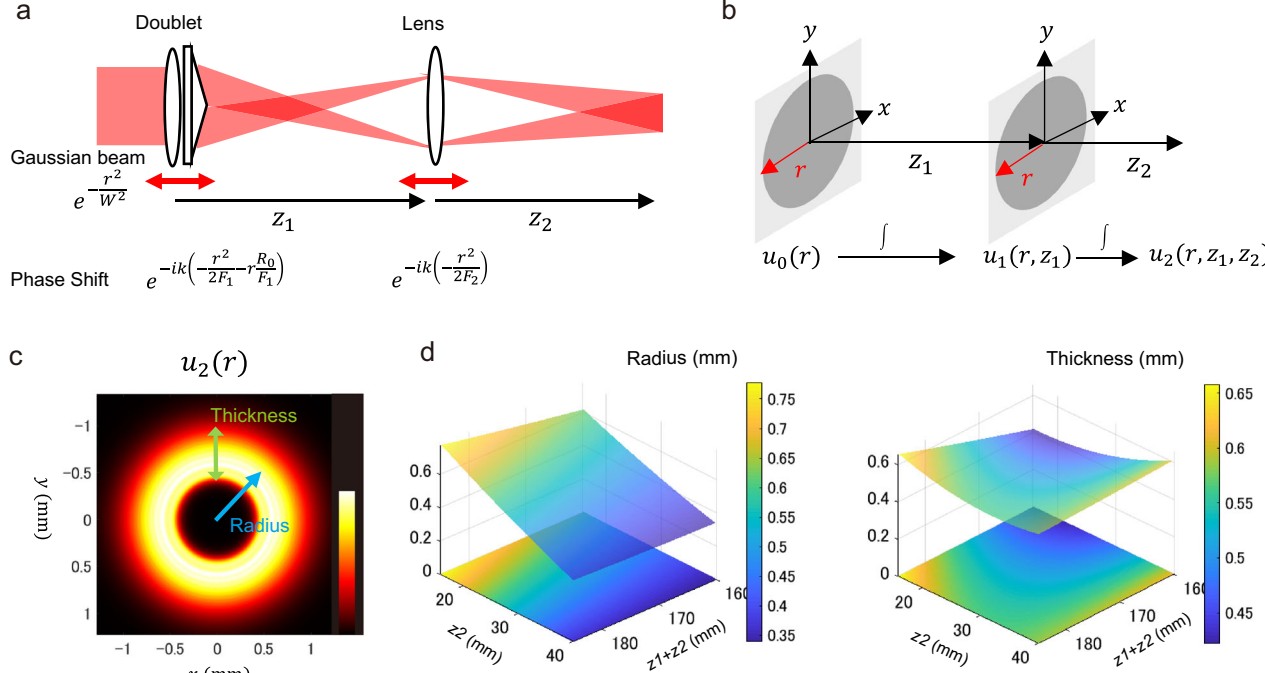

**Fig. 1 Simulation of the lens-axicon triplet. a** Schematic representation of the lens-axicon triplet. Gaussian-shaped laser line is transmitted to the optical unit from left to right. Lens-axicon doublet and a (second) lens can move translationally along the optical axis (red double-headed arrow). Distance between the doublet and the second lens, and that from the second lens are denoted as $z_1$ and $z_2$, respectively. Phase shifts caused by transmission of the doublet and lens are indicated. **b** Illustration of the cylindrical coordinate system. $z$-Direction is set as the optical axis, and the $xy$ plane is set transverse to the $z$-axis. **c** Transverse profiles through the optical unit ($z_1 = 125$ mm and $z_2 = 30$ mm). Parameters (thickness and radius) are measured to characterize the annulus pattern. Radius is defined as the distance between the origin and the maximum intensity point in the radial direction. Thickness is defined as the full width at the half maximum (FWHM) of the distribution in radial direction. **d** Lens and doublet position dependence of the annulus parameters, radius, and thickness estimated by the 2nd order multiple regression model.

imaging more challenging because a larger size requires not only wide FOV but also an increased number of image sequences. These characteristics demand rapid imaging and a lowering phototoxicity in long-term imaging.

Here, we show a simple lens-based illumination optics to stretch the Bessel beam propagation length. To minimize laser energy loss, we attempt to develop a lens-based optical unit that can modify the beam shapes without any devices other than the lenses. A combination of an axicon and two convex lenses can control extension and contraction of the 600–1000-µm full width at a half maximum (FWHM) Bessel beam less than a 4-µm waist without affecting the excitation confinement within the depth of focus of the detection objective (DO) lens. We construct a wide-field two-photon excitation digital scanned light-sheet microscopy (DSLM) using this illumination optics. We achieve ~2 to 3-µm axial resolution when using a 10× NA = 0.3 IO lens, which makes it possible to perform whole-body analysis of medaka development from embryos to juveniles with a cellular resolution. It is further demonstrated that low phototoxicity of the two-photon excitation with Bessel beams enables the long-term time-lapse imaging of embryos over several days. We implement an improvement of the recording mode of DSLM for observing rapid events. This makes it possible to perform 3D time-lapse imaging with over 1 mm³ every 4 s, and demonstrate tracking of inter-cellularly propagating $Ca^{2+}$ wave rounding the embryos. There-fore, our system provides a simple and easy setup for illumination optics suppressing laser powers loss, and it is demonstrated its applicability to DSLM for wide field, fast, and long-term obser-vation of multicellular organisms, broadening the applications of two-photon excitation in light-sheet microscopy.

## Results

**Simulation study of the lens-axicon based beam shaping unit.** To control the shapes of the Bessel beams, we designed a unit called the lens-axicon triplet, which is composed of a lens-axicon doublet and a lens (Fig. 1a). Although the axicon itself creates ring-shaped patterns, the lens-axicon doublet forms the patterns of a Gaussian distributed focal ring[31], resulting in longer extent Bessel beams when projecting the ring beams into the rear pupil of IO lens. Our optical unit further combines the doublet with another convex lens. The system is allowed to translate the lens and lens-axicon doublet along the optical axis, controlling the ring patterns.

We first verified the systems responses of beam profiles against the positions using computer simulations. Numerical integrations of the double Kirchhoff–Fresnel diffraction integral, which emulate the laser beam profiles through the triplet, using the cylindrical coordinate (Fig. 1b) were performed. The intensity distributions in the $xy$-transverse plane show annular patterns (Fig. 1c). We quantified the parameters of the ring, radius $R$, and thickness $T$, and investigated functional forms of the parameters on the plane of $z_1$ and $z_2$. The parameters show different illumination patterns (Fig. 1d). $R$ likely varies linearly in $z_1$ and $z_2$, while the $T$ surface is convex downward in the indicated range of $z_1$ and $z_2$. The different behaviors suggest that an arbitrarily annular beam can be constructed due to tunability of the radius and thickness by adjusting the positions of the doublet and lens.

**Construction of DSLM with the lens-axicon triplet.** We built a DSLM, which possesses a lens-axicon triplet (Fig. 2a). The Gaussian laser beams are transformed through the triplet unit, and the resulting ring shape beams are focused on the Galvano

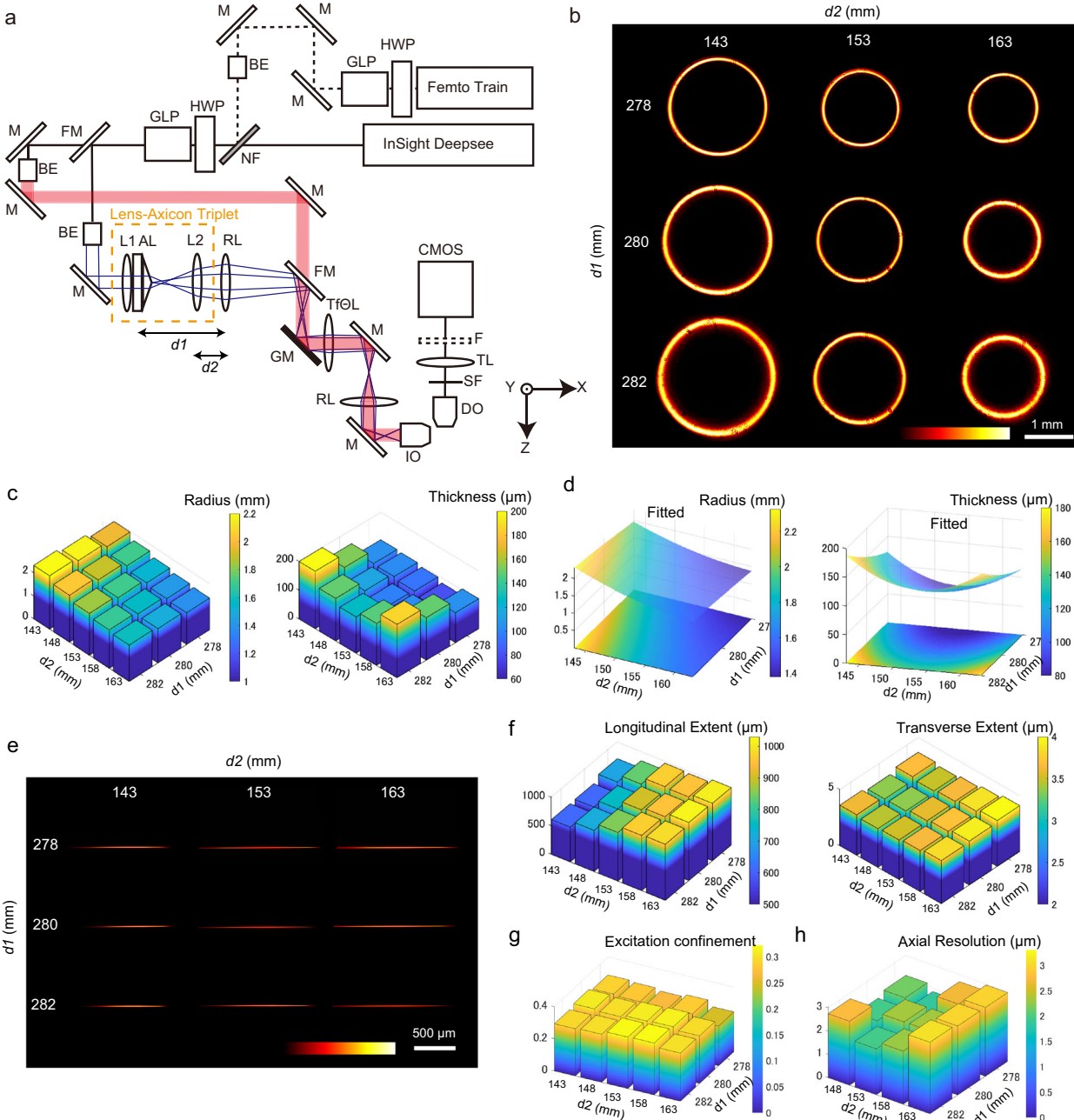

**Fig. 2 Optical properties of the Bessel beam created by the lens-axicon triplet. a** Optical setup of the DSLM system. Two types of illumination optics (the Gaussian and Bessel pathways) are setup. In the Bessel beam system, the lens-axicon triplet in which the doublet and the lens L2 can translate along the optical axis is introduced (orange dotted box). For the illumination source, Semiconductor and Ti:sapphire laser oscillators, Femtotrain (1040 nm wavelength) and Insight Deepsee (700–1300 nm wavelength) are used. The illumination and the detection pathways are perpendicular to each other. Galvano mirror scanner is introduced to generate the light-sheet, and the created optical slice is imaged by the CMOS camera. The sample mounting holder is controlled by the motorized $xyz\theta$ stages. **b** Transverse views of the laser intensity created through the lens-axicon triplet in the Bessel pathway at the back focal plane of IO lens. The dependence of the distance from the relay lens to the doublet ($d_1$) and to lens L2 ($d_2$) on the annulus patterns is shown. **c** Quantified annulus parameters (radius and thickness) for different $d_1$ and $d_2$. **d** Fitted values for the radius and thickness on the plane of $d_1$ and $d_2$. **e** Laser line profiles created by the illumination objective lens for different $d_1$ and $d_2$. Fluorescent signals emitted by the sulforhodamine B solution are imaged through DO lens and the CMOS camera. **f** Quantified line profiles for different $d_1$ and $d_2$. Signal extents (FWHMs) along the longitudinal ($x$) and transverse ($y$) direction at the beam waist are shown. **g** Excitation confinements for different $d_1$ and $d_2$. **e**–**g** For each line profile quantification, a snapshot is analyzed. **h** PSF values based on the fluorescent bead measurements for different $d_1$ and $d_2$. The PSF calculation employed over 90 independent beads as data points. Abbreviations; HWP half wavelength plate, GLP glan-laser polarizers, M mirror, BE beam expander, NF notch filter, FM flip mirror, RL relay lens, AL axicon lens, GM Galvano mirror scanner, TfΘL telecentric fΘ lens, IO illumination objective lens, DO detection objective lens, SF short pass filter, TL tube lens, F emission filter.

mirror scanner, which is placed on the back focal plane of the telecentric fΘ lens. We defined the distance $d_1$ as the distance from the relay lens (fixed position) to the doublet and $d_2$ as that to L2 (Fig. 2a). We measured the ring patterns created through the unit at the back focal plane of the IO lens for different $d_1$ and $d_2$ (Fig. 2b). The quantified ring parameters exhibit focused annular illuminations with a ring radius 1.3–2.3 mm and ring thickness 80–200 μm (Fig. 2c). The system produces parallel beams along the optical axis (Supplementary Fig. 1). Even though an optical mask is not used, sharp and clear annular rings are observed.

The radius $R$ varies linearly in $d_1$ and $d_2$; $R$ decreases as $d_1$ increases, but increases as $d_2$ increases. On the other hand, the $T$ shapes convex downward in the indicated range of $d_1$ and $d_2$, similar to the simulation (Figs. 2c, d). Thus, if an appropriate axis on the $d_1$ and $d_2$ plane is chosen, the ring parameters can be tuned. Along the $d_2$ axis, the variation width of the radius is large, while the thickness along the axis is small. For example, if $d_1 = 280$ mm is fixed, and the radius changes from 1500 μm ($d_2 = 163$ mm) to 2150 μm ($d_2 = 143$ mm), but the thickness is almost constant from 148 μm ($d_2 = 163$ mm) to 152 μm ($d_2 = 143$ mm). Therefore, translating the doublet can vary the radius extensively while maintaining the thickness value.

To determine the two-photon excitation volumes of the Bessel beams, we measured the fluorescence signals emitted by a sulforhodamine solution (Fig. 2e). We further measured the beam profile parameters. The maximum intensity decreases as the longitudinal extent increases (Supplementary Fig. 2). To numerically compare the volume profiles, we estimated FWHM of the beams along the longitudinal and transverse axis at the beam waist. The longitudinal extent varies from 600 to 1000 μm, the transverse extent at the waist is under 4 μm (Fig. 2f), and that at the half-maximum intensity point maintains its narrow values at least in a range of $d_2$ (Supplementary Fig. 3), indicating long and thin excitation profiles. This result suggests that the tunability of the beam propagation length (i.e., FOV) is possible without affecting the beam thickness and, consequently, the in-focus excitation of the DO lens. To evaluate this quantitatively, we evaluated the excitation confinement, which is a measure of the in-focus fluorescence signal strength and is defined as the ratio of the excitation intensity within the depth of focus of the DO lens (2.8 μm) to the total excitation intensity[18]. The result showed that the beams produce ratios of 0.3 and this value is independent of the beam parameters $d_1$ and $d_2$ (Fig. 2g). This confirms that the lens-axicon triplet can control FOV without influencing the optical sectioning capability.

Next, to examine how the ring radius affects the longitudinal extent of the beam, we compared the cases of $d_2 = 163$ mm and $d_2 = 143$ mm ($d_1 = 280$ mm fixed). The former case provides a length of 1024 μm, while the latter gives a length of 614 μm, indicating that a change of the radius (or $d_2$) can control the Bessel beam extent. To investigate the resolution of the microscope, we measured the point spread functions (PSFs) based on fluorescent bead measurements. FWHM values evaluated using whole images show that the resolution along the $z$-axis <3 μm is achieved (Fig. 2h). Then to investigate the positional variability of the resolution, we evaluated the axial resolution at the center and the half-maximum intensity point in the images by creating cropped images of $256 \times 1024$ pixels (Supplementary Fig. 4). The results indicate that both resolutions are ≲3 μm, suggesting that the resolution of the Bessel beam is kept over FOV.

**Comparison of Gaussian and Bessel pathways**. To compare the properties of the Bessel beams with the Gaussian beams, we set

up three types of Gaussian beams, named Gaussian A, B, and C. These Gaussian beams were expanded to a $1/e^2$ diameter of 2.42, 1.69, and 1.42 mm at the back focal plane of the IO lens (Supplementary Fig. 5). They show various line profiles (Fig. 3a). The longitudinal extent of the beams varies from 370 to 810 μm. The extent of Gaussian C is almost equivalent to that of a Bessel beam ($d_1 = 280$ mm and $d_2 = 148$ mm) (Fig. 3b left). The transverse extent at the beam waist becomes broader as the longitudinal extent becomes longer, and those extents are broader than the Bessel beam (Fig. 3b middle). The beam thickness for the Bessel beam is sustained to the half-maximum intensity point of the beam, while those for the Gaussians are expanded (Fig. 3b right). The excitation confinement of the Gaussian beam varies among the three different types (Fig. 3c), while the Bessel beam does not change over the entire range of the longitudinal length (Fig. 2g), clearly indicating an advantage for the use of the Bessel beam in tuning FOV. The results of the PSF measurement using the fluorescent beads also confirm the illumination homogeneity. The estimated PSFs show that the resolution of the Bessel system is higher than all of the Gaussian systems (Fig. 3d). The standard deviation of PSF for the Bessel beam is smaller than that for the Gaussian beams, indicating the variability of the resolution within an image. To confirm this point, we evaluated PSFs at the center and the half-maximum intensity point in the images by creating cropped images of $256 \times 1024$ pixels similar to those for the Bessel systems (Supplementary Fig. 6). The resolution of the Bessel beams has a similar resolution even at the half-maximum intensity point, whereas that for the Gaussian beams assumes a higher value at the half maximum intensity point.

**Application to live imaging of Medaka**. To demonstrate the applicability of a tunable Bessel beam DSLM to bio-imaging, we first investigated the self-reconstruction effect, which is a widely known property of the Bessel beam. This increases the homogeneity of the image with smaller fluctuations. Weakened dark and bright stripes appear behind the obstacles. In fact, it seems that wider shadow lines are recognized in the Gaussian optics, while similar but thinner shadow lines are recognized in the Bessel optics (Fig. 4a). In addition, the fluctuation in the Gaussian is higher than that in the Bessel system. To quantitatively compare image improvements, we extracted the line profiles along the $y$-axis, showing different profiles (Fig. 4a, white dotted lines and Fig. 4b). The intensity standard deviation and the ratio of the higher to lower power spectra of the line profile were used as quantitative measures[13,19]. The standard deviations of both Gaussian beams are significantly higher than that of the Bessel beam, indicating that the lines created by the Bessel beam are relatively flat (Fig. 4c). Next, the ratio of the power spectra of an object smaller than 20 μm to that of the total was calculated using Fourier analysis (Fig. 4d). The power ratio of the Bessel optics is significantly higher than that of the Gaussian A optics, confirming that narrower stripes are created in the Bessel optics. Furthermore, the power ratio of the Gaussian C optics is also higher than that of the Gaussian A optics, suggesting that a broader beam thickness weakens the shadow lines. We then compared images between different Bessel beams, and recognized significant differences in the shadow lines (Fig. 4a, white arrow heads). The standard deviation and power ratio of the Bessel ($d_1 = 280$ mm and $d_2 = 148$ mm) are lower and significantly higher than those of the Bessel ($d_1 = 280$ mm and $d_2 = 158$ mm) beam, respectively (Figs. 4e, f), indicating that the lines are weakened in the shorter extent beam. This indicates that the adaptation of the beam extent to the sample length improves the image quality in terms of the self-reconstruction effect. The embryo size, including

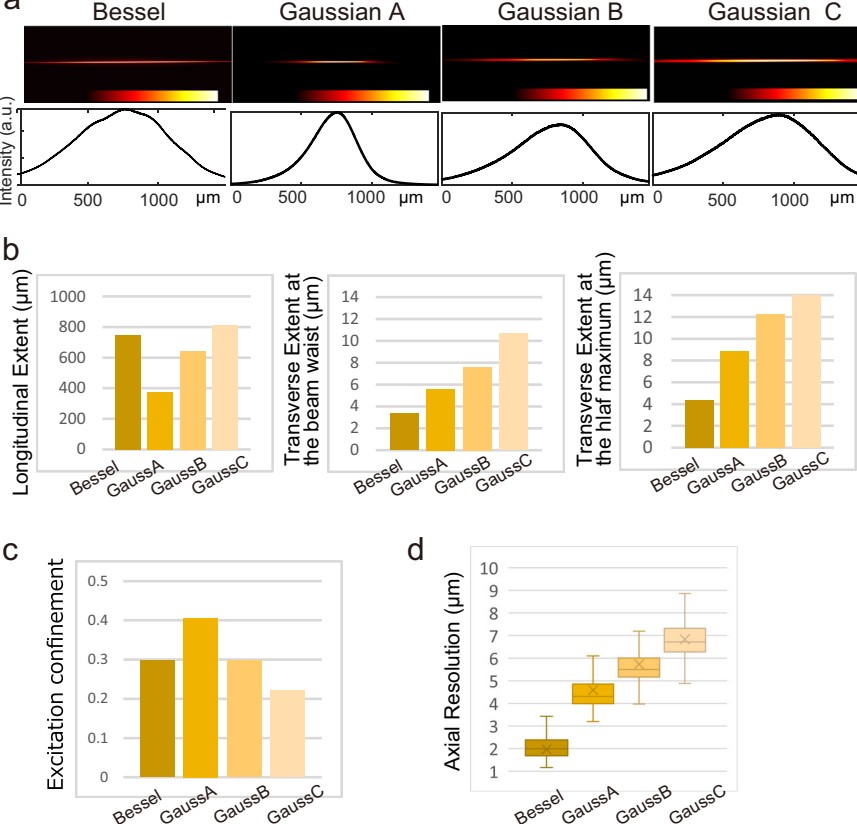

**Fig. 3 Comparisons of the optical properties between the Gaussian and Bessel beams. a** Laser line profiles after transmission of the IO lens for Bessel ($d_1$ = 280 mm and $d_2$ = 148 mm) and three Gaussian beams. Fluorescent signals emitted by the sulforhodamine B solution are imaged. a.u. arbitrary unit. **b** Quantified line profiles. FWHMs along the longitudinal direction (left), transverse direction at the beam waist (middle), and the half-maximum intensity point (right). **c** Excitation confinements. **b**, **c** For each line profile quantification, a snapshot is analyzed. **d** PSF values based on the fluorescent bead measurements. The PSF calculation employed over 90 beads as data points. In the box plots, the box lines indicate the first quantile, median, and third quantile. Lower and upper whiskers indicate the minimum and maximum, respectively. The Cross mark represents the average.

the embryonic body and extraembryonic tissues, is over 1 mm, and the transparency of embryos is higher. Thus, a longer extent beam may be suitable for embryos (Supplementary Fig 7a). On the other hand, for juveniles, due to the increased opacity, it may be better to use shorter extent Bessel beams. Actually, the entire head region of a 2-week post hatching (wph) juvenile can be captured using a longitudinal extent 614 μm ($d_1$ = 280 mm and $d_2$ = 143 mm) beam (Supplementary Fig. 7b). Finally, we measured a 3 wph juvenile to compare the excitation homogeneity inside living tissues between the Bessel and Gaussian systems, which have almost equivalent beam extents. A number of lymphatic endothelial cells at the distal area of the beam incidence can be recognized in the Bessel beam, while only a small population of cells can be recognized in the Gaussian beam (Supplementary Fig. 8, white dashed boxes). This improvement of image contrast for Bessel beams is beneficial in applications to larger biological samples.

To obtain structural insights of the fish, a 2 wph larva of the FLT4-EGFP strain, which expresses EGFP in the lymphatic endothelial cells[32], with angiography by dextran-conjugated tetramethylrhodamine B, was measured (Fig. 5a)[33]. To visualize the whole body of the larva, we acquired tiled four z-stack images where 20–25% of each stack image overlaps. Then using a stitching tool[34], a combined 3D image was constructed. The 3D reconstruction of the image clearly visualizes the whole vascular system of the larva. The blood vessels in red show the morphology of the dorsal aorta, posterior cardinal vein, dorsal

longitudinal anastomotic vessel, intersegmental vein and artery, and the capillary blood vessels in the muscles (Fig. 5a, yellow rectangle). The lymphatic vessels in green clarify the dorsal, lateral, spinal, and intersegmental lymphatic vessels. It is apparent that the lymphatic vessels and blood vessels run alongside each other (Fig. 5a, yellow rectangle), demonstrating cellular resolution imaging. To further describe the anatomical structures of blood vessels in specific organs, we created cropped 3D images in the body trunk, brain, and tail fin (Supplementary Fig. 9). In the body trunk, the dorsal and caudal aorta runs caudally at the center of the body, and the dorsal and lateral intersegmental arteries/veins are recognized (Supplementary Fig. 9a). These vessels bridge the dorsal aorta, posterior cardinal vein, and dorsal longitudinal anastomotic vessel. At the end of the body, the caudal artery branches and enters into the fin. A looping structure of the vessels at the end of the tail was observed (Supplementary Fig. 9b white dotted circle). The parietal vascular system viewed from the dorsal side was also presented (Supplementary Fig. 9c). Although several basic structures, like the pineal gland, anterior veins, and ophthalmic arteries/veins can be observed, the total vascular network is quite complicated. We next captured a whole-brain image of the larvae using the Kif5Aa-EGFP strain, which expresses EGFP in neuronal cells[35]. The major anatomical structures, forebrain, midbrain, and hindbrain are clearly distinguished in the 3D reconstructed image (Fig. 5b, left). Furthermore, in the sliced image, neurons and axons are recognized (Fig. 5b, right, white arrow).

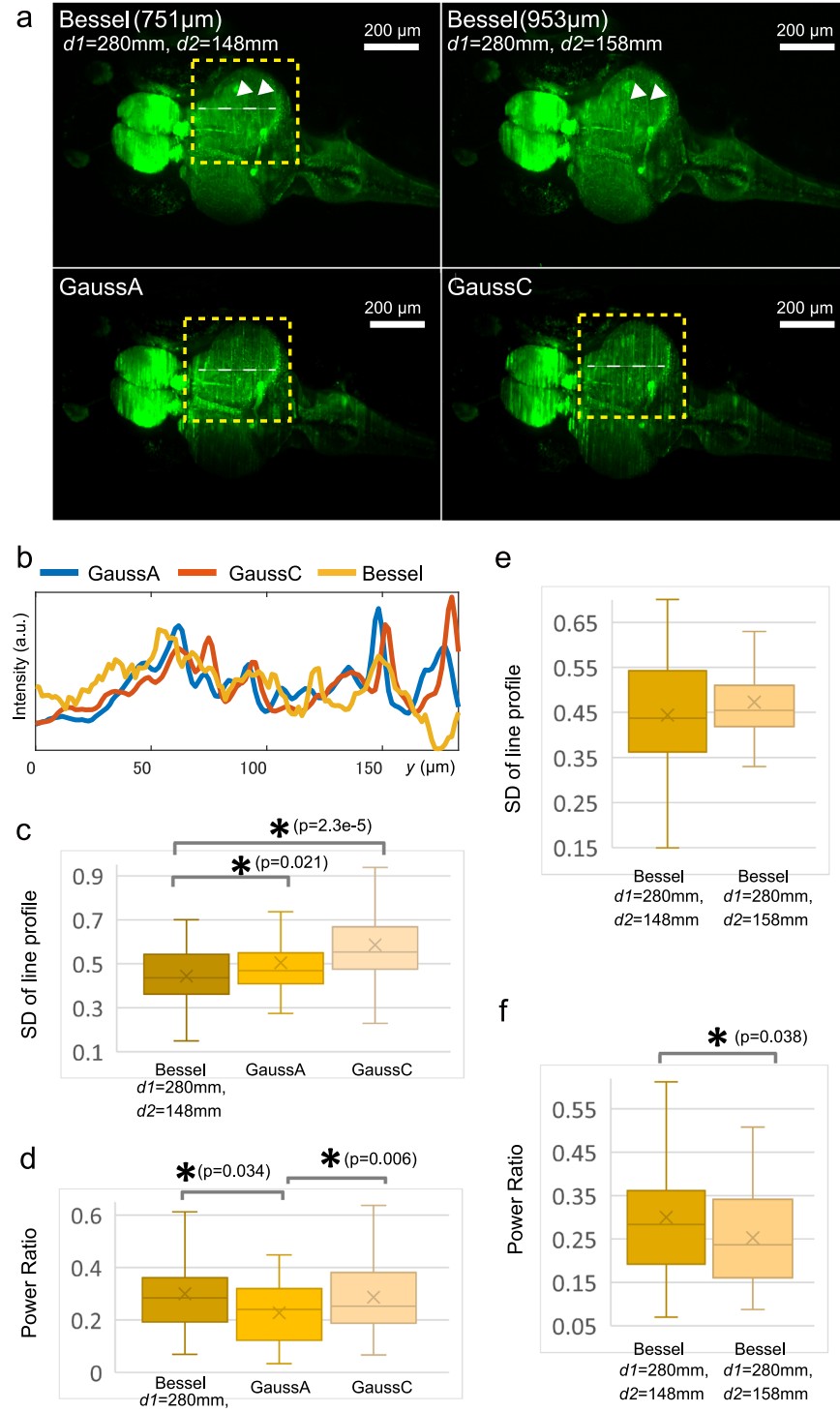

**Fig. 4 Application of the tunable Bessel beam DSLM. a** Comparison of the brain image between the Gaussian and Bessel beams. Larvae of the Kif5Aa-EGFP strain, which expresses EGFP in neuronal cells[35], are used. Images shown are acquired using the same larva. Yellow rectangles indicate the area where significant differences appear in the comparison of Gaussian and Bessel beam irradiations. White arrow heads indicate the area where significant differences appear in the comparison of different Bessel beam irradiations. Scale bar, 200 μm. $n = 5$ biologically independent larvae are subjected to the analysis. **b** Line profiles along the white lines in (**a**). a.u. arbitrary unit. **c** Standard deviations of the line profiles. **d** Ratio of the power spectra of the object smaller than 20 μm to that of the total. **e** Standard deviations of the line profiles for the Bessel beam ($d_1 = 280$ mm and $d_2 = 148$ mm) and ($d_1 = 280$ mm and $d_2 = 158$ mm). **f** Power ratio for the Bessel beam ($d_1 = 280$ mm and $d_2 = 148$ mm) and ($d_1 = 280$ mm and $d_2 = 158$ mm). **c–f** In the box plots, the lines of the box indicate the first quantile, median, and third quantile. Upper and lower whiskers indicate the minimum and maximum, respectively. The Cross mark represents the average. The asterisk indicates a $p$ value <0.05 with one-sided $t$-test. Degrees of freedom for the $t$-test is 29. **c–f** For the analysis, six independent line profiles are extracted from $n = 5$ biologically independent larvae. In total, 30-line profiles are subjected to the analyses.

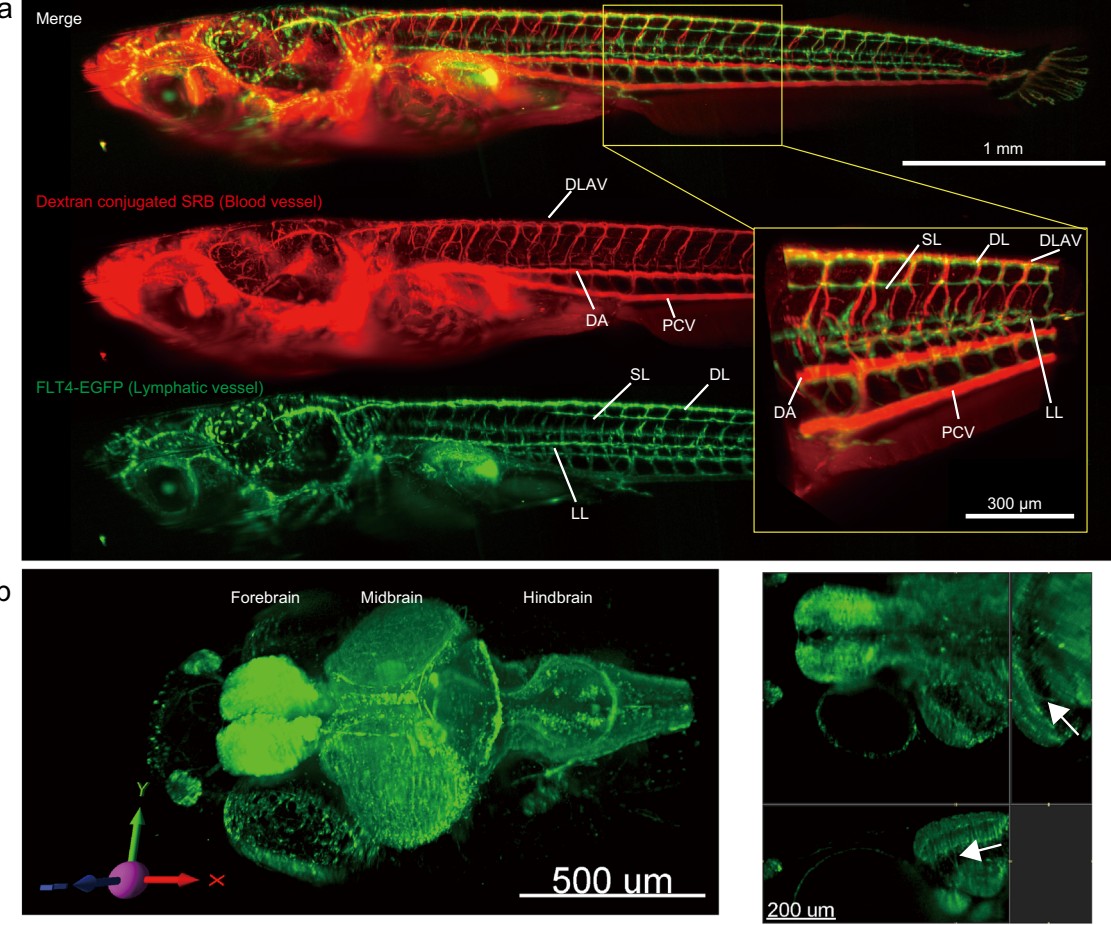

**Fig. 5 Whole-body fluorescent imaging of Medaka larvae. a** Three-dimensional reconstructed view of the lymphatic and blood vessels in the whole body of a 2 wph larva. Green shows the EGFP signals from the FLT4-EGFP transgenic strain, and red shows the dextran-conjugated rhodamine signals from the blood vessel. The yellow rectangular shows a magnification view at the body trunk. Exposure time is 100 ms, z-interval is 2 μm, and xy 1024 × 1024 pixels. n = 3 biologically independent larvae are subjected to the analysis. **b** Three-dimensional reconstructed view (left) and a sliced view (right) of the whole brain of the Kif5Aa-EGFP strain larva. Exposure time, z-interval, and xy are 100 ms, 2 μm, and 1024 × 1024 pixels, respectively. n = 5 biologically independent larvae are subjected to the analysis. Abbreviations; DA dorsal aorta, DL dorsal lymphatic, DLAV dorsal longitudinal anastomotic vessel, LL lateral lymphatic, PCV posterior cardinal vein, SL spinal lymphatic.

**Long-term observation of embryonic development**. We explored how the Gaussian and Bessel beam conditions affect the phototoxicity. To evaluate this, time-lapse 3D imaging of 3 dpf medaka embryos of the β-actin-DsRed2 strain were performed in 5-min intervals (200-ms exposure time/slice, each slice recorded as 2048 × 2048 pixels, recording speed: 2.4 slices/s, 2 μm step of z-sequence) by Gaussian and Bessel beam irradiations. It has been reported that medaka possesses four types of chromatophores: melanophores, xanthophores, leucophores, and iridophores[36]. Melanophores, which contain melanin, are black in color and absorb a wide wavelength range of light. Thus, the absorption of infrared light by this chromatophores can lead to an excessive increase in body temperature, causing photodamage to the embryo. We chose to use 3 dpf embryos for this phototoxicity assessment because the number of the pigmented melanophores largely increases at this stage, especially in the eye[37].

We first examined the survival of the embryos by equalizing the laser average power at 500 mW at the back focal plane of the IO for the Gaussian and the Bessel ($d_1 = 280$ mm, $d_2 = 148$ mm) systems. Embryos died after a single z-sequence of both Gaussian A and C irradiations, while embryos survived after 10- and 20 times z-sequence Bessel irradiation (Table 1). In the dead embryos, the heartbeat stopped and pigment cells in the eyes

were ablated, changing the morphology of the eyes to irregular shapes (Fig. 6a). Moreover, air bubbles near the eye were recognized (Fig. 6a black arrow). Since these beams are generated by injecting lasers of equalized power to the IO, the peak intensities (i.e., photon densities at the peak) differ from each other, which possibly affected the survival results. Thus, we next set up the Gaussian and Bessel beams with equal ability of fluorescence excitation. We determined that an average laser power of Gaussian C optics of 325 mW generated an equal intensity with a Bessel beam ($d_1 = 280$ mm, $d_2 = 148$ mm) of 500 mW (Supplementary Fig. 10). With this Gaussian beam setup, all embryos died after ten times z-sequence measurements (Table 1). These results suggest that peak photon density and the beam thickness are both critical to the survival because the Gaussian beam is thicker than that of the Bessel beam when equalizing the longitudinal beam extent. The fluorescent 3D images indicate that the ablated pigment cells emit strong fluorescent signals (Fig. 6b white arrow). These ablations should be due to the absorption of infrared light by the pigment cells. These findings are supported by the experiments using the see-through-based FLT4-EGFP strain where pigmentation is greatly reduced (Table 1).

To further reduce the photodamage, we implemented a non-stop recording mode, which decreases the recording time and the

**Table 1 Survival of embryos in phototoxicity experiments.**

| Medaka strain | #Time-lapse recording | Bessel 500 mW | | Gaussian C 325 mW | GaussianC 500 mW | | Gaussian A 500 mW | |
|---|---|---|---|---|---|---|---|---|
| | | **#Alive** | **#Dead** | **#Alive** | **#Dead** | **#Alive** | **#Dead** | **#Alive** |
| DsRed2 | ×1 | 3 | 0 | | | 0 | 3 | 0 |
| | ×10 | 3 | 0 | 0 | 3 | | | |
| | ×20 | 3 | 0 | | | | | |
| FLT4 | ×10 | 3 | 0 | | | 3 | 0 | |

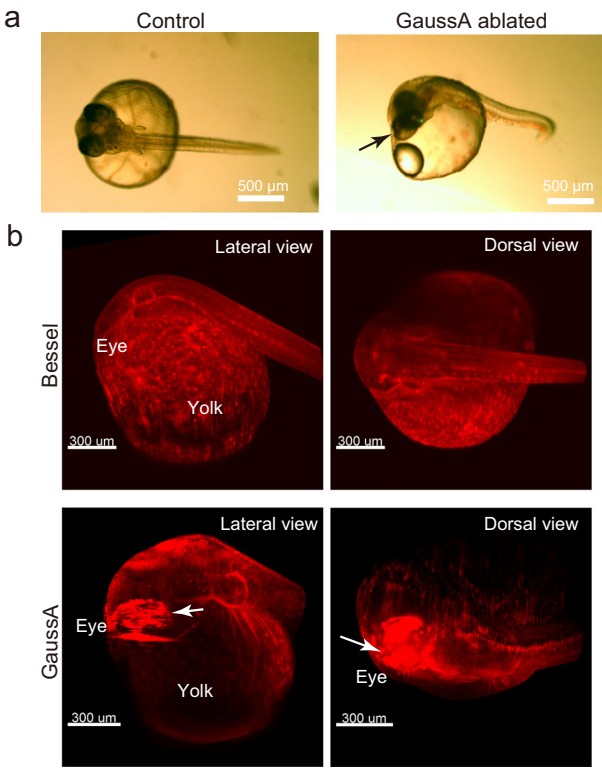

**Fig. 6 Phototoxicity assessment. a** Bright field view of the 3 dpf embryos of the β-actin-DsRed2 strain after the phototoxicity experiment. Control indicates a gel-embedded embryo, which does not show significant damage. For the ablation experiment, the time-course 4D imaging was performed in 5-min intervals with 200 ms exposure/slice with 2 μm steps, 2048 × 2048 pixels, ~600 slices. The arrow shows the air bubble created around the eye. **b** 3D view of the embryos viewed from the lateral and dorsal side. White arrows indicate strong fluorescence emitted from the retinal pigments. **a**, **b** n = 3 biologically independent β-actin-DsRed2 strain embryos are examined for the different beam conditions of phototoxicity experiment.

laser beam radiation to the sample. Hence, we were able to perform long-term live tracking of embryos. In this mode, the motorized stage moves continuously at a constant speed and a camera records every set of the exposure time. To evaluate the resolution, we estimated the PSF for both the stop and non-stop mode. It is 2.47 ± 0.18 μm (average±standard deviation) for the stop mode and 2.05 ± 0.45 μm for the non-stop mode. We measured time cost for recording (100 ms exposure time, 1-μm z-stack interval, total 1-mm z-range). The measured period for the non-stop mode is 2.6 times faster than that for the stop mode (416 s for the stop and 162 s for the non-stop mode).

Using this mode, we performed longer term (several days) time-lapse tracking of lymphatic vessel formation[33]. We measured an embryo of the FLT4-EGFP strain from ~80 hpf, the early stage of lymphangiogenesis, to the hatching stage (~160 hpf). Every 5 min, we took a 3D volume image with a 2-μm stepping size (100 ms/slice). The reconstructed images depict the process of lymphatic vessel formation (Fig. 7a). At 84 hpf, the lymphatic endothelial cells start the initial formation of the lymphatic vessel at the body trunk. At 122 and 160 hpf, the EGFP is strongly expressed in the whole body, and tube formation is observed. To perform detailed analysis of how the lymphatic vessel develops, we focused on the dorsal lymphatic formation (Fig. 7b). The cells forming the dorsal lymphatic migrate posteriorly in a straightforward way. These migrating cells connect the intersegmental lymphatics (Fig. 7b). In addition, in the most posterior region, the migration mode appears to differ, namely, the cells from the portion of intersegmental lymphatics sprout, elongate cranio-caudally, and connect with each other to form the lymphatic (Supplementary Movies 1 and 2). These observations demonstrate the power of whole stage analysis to unveil developmental dynamics.

**Fast 3D time-lapse imaging captured intercellularly propagating $Ca^{2+}$ wave.** To track more rapid biological events such as $Ca^{2+}$ signals, we further formulated a new recording mode: bidirectional recording. The bidirectional recording mode performs onward-and-return recording along the z- axis, decreasing the re-positioning time of the motorized stage to the origin (Fig. 8a). To demonstrate fast 3D time-lapse imaging with these modes, we analyzed 3D propagating $Ca^{2+}$ oscillations. During embryonic development of Medaka, rhythmic contraction waves propagate across the out-most layer of extraembryonic tissues, which cover both the developing embryo and the yolk cell. This rhythmic contracting movement is responsible for intracellular elevation and intercellular propagation of $Ca^{2+}$[38].

To catch this event, a combination of the above described two recording modes were used. Hence, we performed time-lapse analysis of this $Ca^{2+}$ wave with a genetically encoded fluorescent $Ca^{2+}$ indicator GCaMP6f[33]. GCaMP6f expressing embryos were measured in 20-μm steps (20 ms/slice) with 4-s time intervals over a total of 30 min. The fluorescent signals are detected at the surface of embryos (Fig. 8b, Supplementary Movie 3). To visualize this dynamic process easily, we performed a planar projection of the spherical shell data using a coordinate transformation and a dimensionality reduction technique (Supplementary Fig. 11). We transformed the Cartesian coordinate to the polar coordinate system where the centroid of embryo was set to the origin of the coordinate system. Using a morphological feature in which the embryonic shape can be approximated as a sphere, a 3D image was reduced to a 2D image without loss of important information. Thus, maximum intensity projection with respect to radius shell $r \sim r + \Delta r$ was performed to visualize the activity on

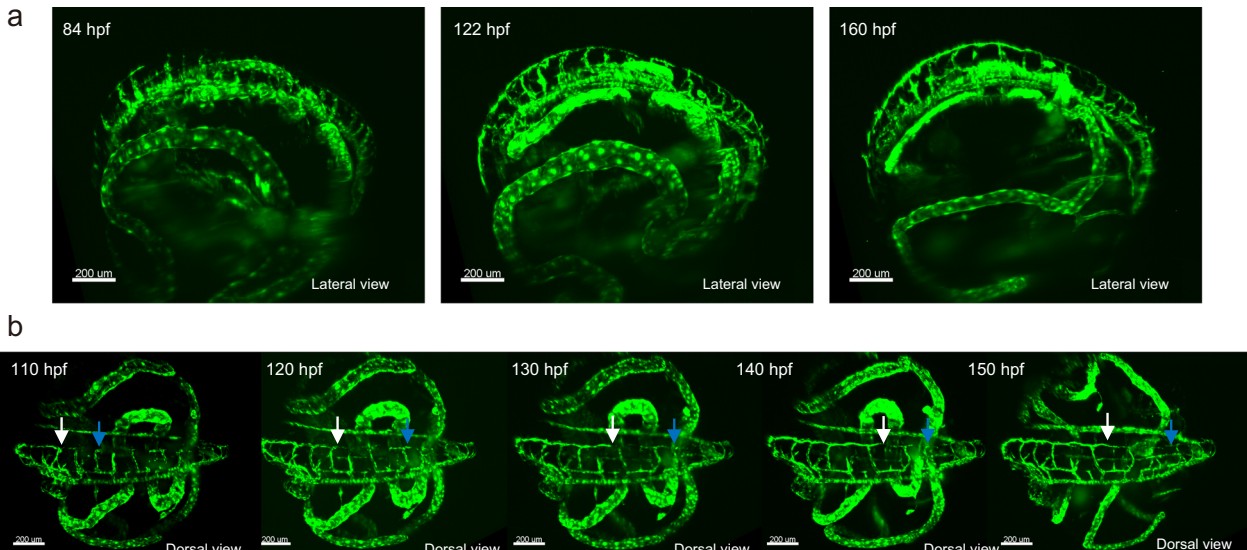

**Fig. 7 Long-term time-lapse imaging of developing embryos. a** Snapshot images of lymphatic vessel development in a FLT4-GFP embryo for stages of 84, 122, and 160 hpf. Embryo is embedded into 1% hollow agarose gel filled with 0.03% seawater. **b** Detailed time-lapse observation of the lymphatic endothelial cell migration. White arrows indicate migration front, while blue arrows indicate migrating cells from the intersegmental lymphatics. (**a**, **b**) $n = 2$ biologically independent embryos are subjected to the analysis.

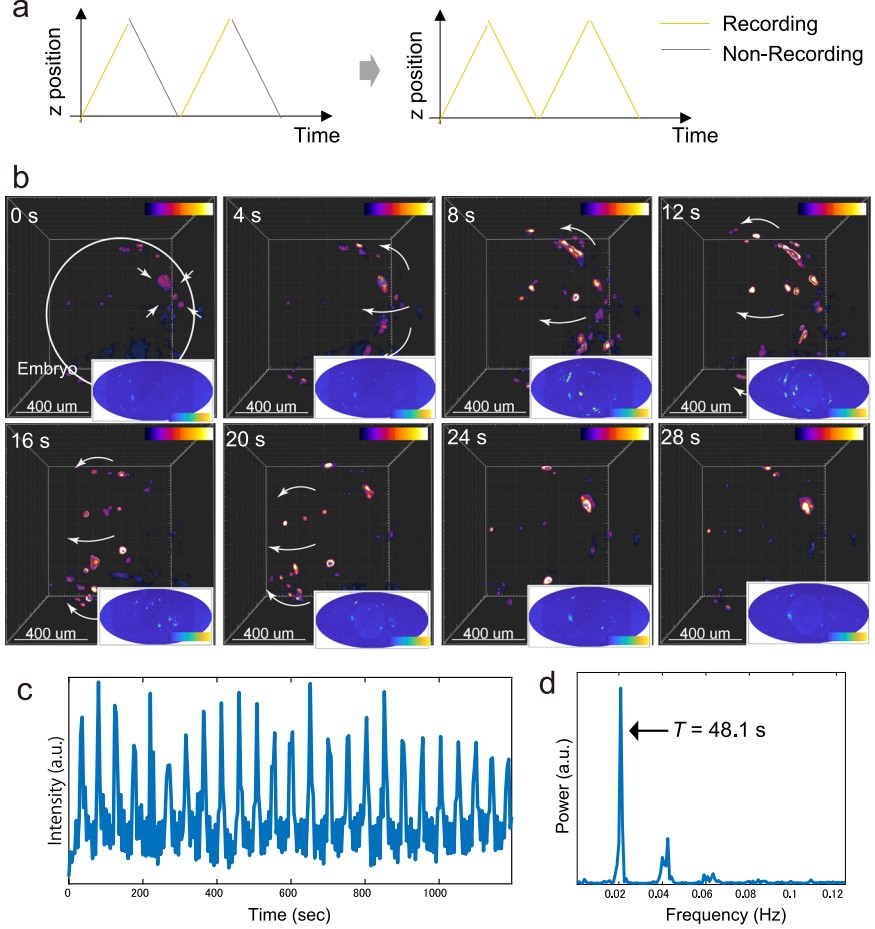

**Fig. 8 Fast 4D imaging of intercellular propagating Ca$^{2+}$ wave. a** Change of the recording mode. To-and-from bidirectional recording mode are implemented. **b** 4D time-lapse imaging of a GCaMp6f expressed Medaka embryo (1 dpf). Fluorescent signals propagating over the outer layer surrounding whole embryos are indicated. Bottom-right images are two-dimensionally projected maps of the signal using the Mollweide cartography technique. **c** Plot of the GCaMP6f signal in a small ROI as a function of time. a.u. arbitrary unit. **d** Fourier analysis of the oscillatory expression of the GCaMP6f signal. a.u., arbitrary unit.

the plane. Finally, the voxels with $\theta\varphi$ were mapped into the elliptic plane by the Mollweide projection method (Fig. 8b right bottom frame). An initial elevation of the signal in a single-cell or small population of cells (Fig. 8b 0 s white arrows) occurs, and this signal transmits to neighboring cells. The transmission pattern seems to be isotropic. Namely, the $Ca^{2+}$ signal radially expands and then the wave propagation proceeds globally (Fig. 8b 4–16s white arrows). Finally, the wave front likely reaches the opposite side of the initiation point and the wave is terminated (Fig. 8b 24 s). The velocity of the traveling wave can be estimated roughly as ~60 μm/s (half perimeter of embryo/traveling time), which is consistent with the previously reported result[38]. The wave restarts after a short interval, and this occurs repeatedly with the same propagation pattern (Supplementary Movie 3). Since the $Ca^{2+}$ signal induces cell contraction, the position of the cell changes over time, indicating that the contraction movement follows the $Ca^{2+}$ signal. To quantify the period of the $Ca^{2+}$ wave, we plotted the signal intensity changes with respect to time by selecting ROIs in the images (Fig. 8c). The period seems stable. Figure 8d shows the power spectrum of the signal after a Fourier transformation. The sharp peak means stable periodic patterns with a period of 48 s. Therefore, using a high-speed recording mode, microscopy allows 4D fast imaging of ~1.5 mm size with a 4-s time interval to be performed, despite the compromised axial resolution (20-μm step of the z-sequence).

## Discussion

This study investigates simple lens-based illumination optics for stretching of Bessel beam length, and demonstrates a wide-field two-photon excitation DSLM for fast, gentle, and long-term imaging of multicellular organisms. DSLM achieves 600–1000-μm FOV with ~2–3-μm axial resolution when using 10× NA = 0.3 IO lenses. This allows whole-body imaging of medaka juveniles with a cellular resolution. Because our system offers a simple and easy setup for a tunable illumination optical system, it can be a useful tool to analyze a wide variety of living organisms.

The lens-axicon triplet is composed only of an axicon lens and two convex lenses. One merit of using this lens-based optics to create non-diffracting beams is that the energy loss can be suppressed. Spatial light modulator can easily tune the beam patterns, but using this device results in laser energy loss. The energy loss due to optical masking is also not negligible. Especially, for a two-photon excitation, the laser power is critical for efficient fluorophore excitation. Thus, it influences the signal-to-noise ratio of the acquired image. To avoid laser energy loss, the lens-based equipment is kept as simple as possible. From the official information of Thorlabs, the reflectance at a 900-nm wavelength by axicon (B-coat) is 0.3% and lenses are 0.4%. The estimated energy loss through the unit is about 1%, indicating a high transmission rate. Keeping this high transmission rate, our optical setup realizes a sharp and clear annular ring beam without using these devices. Adjustable properties of the annulus patterns and the resulting Bessel beam profiles by the unit are confirmed numerically and experimentally. The functional characteristics of quantitative annulus parameters, the radius and thickness, depend on the lens position. The radius can vary greatly compared with the thickness, demonstrating tunability of the radius while keeping a similar thickness by adjusting the positions of the doublet. These patterns affect the propagation length of the Bessel beams. The results of the Bessel beam profiles show that the longitudinal extent varies from 600 to 1000 μm, while the transverse extent is kept under 4 μm. The tunability is ensured by the excitation confinement, which is a ratiometric measure of the in-focus to the total excitation signals. The ratio of the Bessel beams is around 0.3 over the entire range of the propagation length,

while those of the Gaussian beam vary as the length changes. This shows the constant optical sectioning capability of the Bessel beams created by the proposed optical unit. These results suggest that the central lobe thickness of the Bessel beams remains nearly unchanged even though there may be changes in the energy balance between the central and side lobes of the Bessel beam, which affect the propagation length but the fluorescence from the side lobe is less intense due to two-photon excitation.

This is achieved using ×10 magnification with NA = 0.3 objective lens for illumination. Previous reports of two-photon Bessel beam illumination tended to use higher NA objective lenses, NA = 0.8[17], NA = 0.3[21], NA = 0.8[19], and NA = 0.42[27]. The use of a slightly lower NA objective lens also contributes to the laser beam extension. Our DSLM is the first report of a two-photon excitation DSLM possessing over 1-mm FOV. The previously reported widest FOV using the two-photon Bessel beam is 600 μm[26]. However, the FOV is determined by a combination of IO and DO lenses. Thus, this wide FOV does not express pure ability of the microscopy system compared with the previous ones. The beam profiles and PSF estimation based on fluorescence bead measurements indicate that the Bessel beam illumination is more uniform over the entire FOV than the Gaussian illumination.

Applications of the tunable DSLM demonstrated imaging analysis of medaka development from embryos to juveniles in a single scheme by adapting the beam extent to the samples. A previous study[39] reported length-adaptive Bessel beams in which the focal position and beam length of Bessel beams are automatically adapted to the samples by modulating SLM. This approach can cover samples lying over the entire FOV, but only applies to thin samples that do not fill the whole FOV. Our Bessel beams cover large FOVs while maintaining similar spatial resolutions. Hence, our system is applicable to larger-size samples. We confirmed the self-reconstructing effect in in vivo imaging of Medaka, reducing the appearance of shadows behind objects. This means that the Bessel beam self-reconstructing effect can improve the image quality despite using the low NA IO lens. Furthermore, the lower phototoxicity of the two-photon Bessel beam was demonstrated by a survival assessment of embryos. Comparison between Bessel and Gaussian beams with equal beam extents and peak intensities indicated a higher survival of embryos in the Bessel beam. A major difference between the profiles of the Bessel and Gaussian beams is the thickness of the main lobe due to suppression of the side lobe fluorescence excitation in Bessel beams in a two-photon excitation. Therefore, the two-photon excitation Bessel beam is less phototoxic than the Gaussian beam because thinner beams can be created in Bessel systems.

An anatomical characterization of the blood vascular system of medaka for adult fish and embryos has been performed[40,41]. Our result is the first report of whole-body blood vascular imaging of medaka larvae (2 wph), but its anatomical features can be partly compared with the previous results. From the body trunk to tail, the morphological characteristics are basically consistent. Observations of the dorsal aorta, posterior cardinal vein, intersegmental arteries/veins and dorsal longitudinal vessel, and the number of vessels branched from the caudal artery at the fin are consistent with previous findings. In[40], it was reported that an irregular alternative appearance of intersegmental arteries and veins occurs. In our results, the intersegmental vessels can be recognized, but it is difficult to distinguish whether they are an artery or a vein since these are morphologically similar at this stage. To further clarify this point, we may need to track later stages of the larvae or construct the transgenic strains which differentiate arteries and veins with different colors. The only difference we observed is that we could not observe the vascular

plexus of the spinal cord located above the vertebrae as described in[40]. In the brain, several basic structures are recognized, but it is too complicated to annotate more specific structures. There may be differences between the vascular anatomy of the embryos and the larvae, but how these differences appear is not clear. Our analysis used larvae while the reported ones used adult fish and embryos visualized with different imaging modalities, scanning electron microscopy, and macroscopic optical imaging. Therefore, to clarify the above points, a detailed whole body and whole developmental stage analysis of the vascular system based on light-sheet microscopy should provide interesting future directions for study.

We improved the recording mode of DSLM for high-speed 3D time-lapse imaging. Recording of 3D images is usually performed by scanning $xy$-plane images for each $z$-position by moving a motorized stage along the $z$-axis. The rate-limiting step for this 3D recording is stage positioning because it is time-consuming to return to the original $z$-position after recording is performed. To reduce imaging time, two recording modes, the non-stop and bidirectional recording, are implemented. When these modes are applied, the motorized stage keeps moving while recording and onward-and-return recording along the $z$-axis, which drastically reduces time. The measured time for the non-stop mode is 2.6 times faster than that for the stop mode. Furthermore, the estimation of the PSF indicates that the spatial resolution does not differ significantly between the stop and non-stop mode. Hence, embryos are measured successfully over 1.2-mm every 4 s.

We attempted to capture intercellularly propagating Ca$^{2+}$ wave rounding the outer layer of embryos. This wave is responsible for contraction of stellate cells[38]. This wave occurs with a period of 40–60 s, and a propagating wave rounds whole embryos within 30 s. To observe these Ca$^{2+}$-dependent contraction waves, we implemented a gel cage sample mounting method (Supplementary Fig. 12), in which the 1% agarose gel mounted with the sample holder was used to make a hole, and then an embryo was placed with medium, finally the chamber was capped by the gel. Mounting biological samples with stiff agarose gel may inhibit cellular movements. Hence, we employed a gel cage method, and successfully captured the cellular contraction movement Ca$^{2+}$ dynamics accompanying the morphological dynamics.

Visualization and understanding of 3D volume data remain challenging. Visualization of 3D images usually relies on the volume-rendering method, which displays a 2D projection of 3D volume data. For a single time point data, digital rotation of the sample image compensates for the 3D information. However, for multi time point data, it is difficult to see an image from many angles simultaneously. Hence, a time series movie with a single angle projection may cause information loss. An efficient way to process 3D data without loss of information is to exploit the geometry of organisms[42]. We implemented a 2D spherical projection by taking advantage of the spherical shape of embryos so that the Ca$^{2+}$ wave just propagates on the embryo surface. By computing the maximum intensity projection with respect to the radius shell, the activity can be mapped onto the sphere. Furthermore, we used a cartography technique, the Mollweide projection method, to visualize the entire tissue in a single 2D image. This clearly visualized cell-to-cell wave propagation, making it easier to estimate the oscillation frequency as 48 s. The detailed dynamics of this wave propagation remain to be elucidated. It would be interesting to study the system properties of this wave using mathematical models. Phenomenological models, which are useful to extract hidden spatiotemporal dynamics[43] and detailed mechanism-oriented models for Ca$^{2+}$ signal[44], are both essential to uncover the dynamics of wave initiation, propagation, and termination.

## Methods

**Optical systems.** Figure 2a shows the optical setup of DSLM. The $xy$-plane was set to the image plane of the DO lens, while the z-axis was orthogonal to that plane. Two femtosecond pulsed infrared lasers, InSight DeepSee (Spectraphysics) and Femtotrain (Spectraphysics), were used for the excitation beam, which propagated along the x-axis, and the light sheet was generated in the xy-plane by scanning the beam along the y-direction using a galvano mirror scanner (Model 6200H, Cambridge Technology). After the Galvano mirror scanner, a telecentric fΘ lens was introduced such that the tilted laser lines were converted into vertical lines. The Insight Deepsee emitted pulses with a duration of 120 fs, a repetition rate of 80 MHz over the near-infrared wavelength range of 700–1300 nm, and working with the average power 1.4 W at 925-nm wavelength. The Femtotrain emitted pulses with a duration of 370 fs, a repetition rate of 10 MHz, and working with the average power 3.5 W at 1040-nm wavelength. These two lasers were combined into the optical pathway through a notch filter of the stop wavelength of 1050 nm.

The Gaussian and Bessel beam pathways were switched by a flip mirror. In the Bessel beam pathway, the incident Gaussian beam was expanded to a $1/e^2$ diameter of 5.74 mm using a beam expander and transmitted to the lens-axicon triplet to create an annulus ring pattern. The lens-axicon triplet, which was composed of a lens-axicon doublet and a lens, was designed to tune the annulus patterns, allowing the longitudinal beam extent to be modified. Details of the unit's properties are described below. In both the Gaussian and the Bessel pathways, the laser powers of Insight DeepSee were tuned using HWP, GLP was placed before the branch of the Gaussian and Bessel systems (Fig. 2a), and the average laser powers were measured at the back focal plane of the IO lens.

The objective lenses used were ×10 magnification dry lens (LMPLN10XIR, ×10 magnification, NA = 0.3, Olympus) for illumination and ×10 magnification multi-immersion lens with a long working distance (CFI-PlanApo 10XC Glyc, ×10 magnification, NA = 0.5, Nikon) for detection.

To create the light sheet, the laser lines were scanned by the Galvano mirror scanner at a rate of 100 Hz along the y-axis. This laser-induced light sheet was imaged through the DO lens, a tube lens with a 180-mm focal length, and a CMOS camera with a 6.45-μm pixel size with the same horizontal and vertical aspect ratio, total 2048 × 2048 pixels (Orca flash 4.0 v3, Hamamatsu Photonics). The images were recorded as 16-bit gray-level images. During the emission pathway, a short path filter at 700 nm (Semrock) was inserted to cut the excitation lights, and the band pass filters at 520/35 and 593/46 nm (center wavelength/bandwidth) (Semrock) were used for multi-color imaging. These emission filters were switched using the motorized filter wheel (Thorlabs). The hand-built sample chamber placed between the IO and DO lenses was made from acrylic plates and cover glasses of 0.12–0.17 mm thicknesses (Matsunami). The sample mounting holder was placed on the motorized stages (M-111.1DG and M-116.DG, Physik Instrumente), which controlled the translation in the $xyz$-directions and the angle rotation in θ along the y-axis. The z-stack image sequence was captured by moving the sample in the z-direction.

A software program to control the camera, motorized stages, motorized filter wheel, and Galvano mirror scanner was made with Labview2015 software (National Instruments). For fast 3D imaging, we set up two recording modes, non-stop recording and to-and-from bidirectional recording. Non-stop means that the motorized stage kept moving at a constant speed and image capture was continuously performed. To-and-from bidirectional recording performed bidirectional recording along the z-axis (Fig. 6a). The image data was initially stored in the random access memory. Then after the z-stack data were recorded, the data were readout to the hard disk drive.

**Simulation and experimental studies of the lens-axicon triplet.** The lens-axicon triplet was a combination of a lens-axicon doublet, which could create a thin annulus pattern with a Gaussian distributed focal ring[31] and a lens. In our setup, the lens-axicon doublet was composed of an axicon with a physical angle $\alpha = 0.5°$ and a convex lens with a focal length of 150 mm. The second convex-lens had a 50-mm focal length. The tunability of the parameters of the annulus pattern, radius, and thickness, was verified by adjusting the positions of the doublet ($z_1$) and lens ($z_2$) (Fig. 1A). To evaluate the beam profiles, we used the Kirchhoff–Fresnel diffraction integral in the cylindrical coordinate system, which is given as

$$u_1(r, z) = \frac{k}{z} \int_0^a u_0(\rho) e^{-ik\frac{\rho^2}{2z}} J_0\left(\frac{kr}{z}\rho\right) \rho d\rho, \qquad (1)$$

where $\rho$ is the radial coordinate, $J_0(\rho)$ is the 0th order Bessel function of the first kind, $a$ is the aperture radius, and $k$ is wave number, $k = \frac{2\pi}{\lambda}$. The amplitude distribution $u_0(r, z = 0)$ along the z-axis is transmitted to $u_1(r, z)$ through the integral. Since the unit considered has two phase shifts, a double integral is required

$$u_1(\rho, z_1) = \frac{k}{z_1} e^{ik\frac{\rho^2}{2F_2}} \int_0^a u_0(\sigma) e^{-ik\frac{\sigma^2}{2z_1}} J_0\left(\frac{k\rho}{z_1}\sigma\right) \sigma d\sigma, \qquad (2)$$

$$u_2(r, z_2, z_1) = \frac{k}{z_2} \int_0^a u_1(\rho, z_1) e^{-ik\frac{\rho^2}{2z_2}} J_0\left(\frac{kr}{z_2}\rho\right) \rho d\rho, \qquad (3)$$

where the term $e^{ik\frac{\rho^2}{2F_2}}$ is the phase shift by the second lens. The amplitude distribution $u_0$ is given by the product of the incident Gaussian beam distribution, the phase retardation of the lens, and the axicon. Thus, $u_0(r) = \exp\{-\frac{r^2}{W^2} - ik(\frac{r^2}{2F_1} + r\frac{R_0}{F_1})\}$,

where $W$ is the Gaussian beam waist and $F_1$ is the focal length of the lens. The radius of the focal ring $R_0$ generated by the doublet is given by $R_0 = (n - 1)\alpha F_1$, where $n$ is the refractive index of the axicon. The parameters $d_1$ and $d_2$ are related to $z_1$ and $z_2$ as, $d_1 = z_1 + z_2$ and $d_2 = z_2$.

We numerically investigated this integral using the trapezoidal rule with an integration step of $\delta = 0.004$ mm. The intensity distributions, $I = |u_2|^2$, in the $xy$-plane for arbitrary $z$ were calculated. The MATLAB program codes that perform numerical integrations were provided as Supplementary Code. Different lens positions showed different annulus patterns. To investigate the dependence of the lens positions on the annulus parameters, we quantified ring radius, $R$, which was defined as the distance between the origin and the point of maximum intensity value in the radial direction, and ring thickness, $T$, which was defined as FWHM of the distribution in radial direction.

For statistical analysis of the function form, we used a nonlinear multiple regression model of the 2nd order,

$$R = \alpha_{00} + \alpha_{10}d_1 + \alpha_{01}d_2 + \alpha_{20}d_1^2 + \alpha_{11}d_1d_2 + \alpha_{02}d_2^2, \quad (4)$$

$$T = \beta_{00} + \beta_{10}d_1 + \beta_{01}d_2 + \beta_{20}d_1^2 + \beta_{11}d_1d_2 + \beta_{02}d_2^2, \quad (5)$$

where $\alpha_{ii}$ and $\beta_{ii}$ were the regression coefficients. These 2nd order expressions were well fitted because the determination coefficients R2 were calculated R2 = 0.981 for $R$, and R2 = 0.998 for $T$. The estimated values of coefficients were $\alpha_{00} = -0.234$, $\alpha_{10} = 0.18$, $\alpha_{01} = 0.0044$, $\alpha_{20} = 0.00011$, $\alpha_{11} = -0.00022$, $\alpha_{02} = 1.57$, $\beta_{00} = -1.64$, $\beta_{10} = 0.058$, $\beta_{01} = 0.011$, $\beta_{20} = 0.00036$, $\beta_{11} = -0.00044$, $\beta_{02} = 1.59$. The radius $R$ and the thickness $T$ were plotted as functions of $d_1$ and $d_2$. $R$ was a monotonically increasing function of $d_1$ and $d_2$ since $\frac{dR}{dd_1}$ and $\frac{dR}{dd_2}$ are always positive and negative in the indicated range of $d_1$ and $d_2$, respectively. On the other hand, $T$ has extrema in $d_2$, since $\frac{dT}{dd_2} = 0$ exists.

In the experimental setup, although the beam propagated through additional lenses (convex lenses and telecentric fΘ lens), the properties described above were transmitted. We measured the beam profiles at the back plane of the IO lens using a beam profiler (SP620U, Ophir-Spiricon) with a software program (Beam Gage Professional6.12, Ophir-Spiricon). The acquired image was outputted as $1601 \times 1200$ pixels image (resolution, 4.4 μm/pixel). To numerically evaluate the Bessel beam shapes, we estimated the annulus parameters of the center, radius, and thickness of the ring by fitting the annulus function $f(x,y) = \exp\left(-(r - r_0)^2/\sigma_0^2\right)$, where $r = \sqrt{(x - x_0)^2 + (y - y_0)^2}$ with the images. We used the radius and FWHM of the ring thickness ($2\sigma_0\sqrt{\ln 2}$) as measures for beam shape. The pattern depended on $d_1$ and $d_2$ (Fig. 2).

As in the analysis above, we performed 2nd order multiple regression analysis. These were well fitted (R2 = 0.999 for $R$, and R2 = 0.938 for $T$, $\alpha_{00} = -418.56$, $\alpha_{10} = 0.14$, $\alpha_{01} = 2.88$, $\alpha_{20} = 0.0012$, $\alpha_{11} = -0.0002$, $\alpha_{02} = -0.0045$, $\beta_{00} = 1.31$, $\beta_{10} = -294.95$, $\beta_{01} = -791.81$, $\beta_{20} = 0.58$, $\beta_{11} = -0.43$, $\beta_{02} = 1.33$.). The results also showed that $R$ is a monotonical function, and $T$ has extrema in $d_2$.

**Evaluation of the DSLM optical properties.** To measure the beam profiles at the focal plane of DO lens, the sample chamber was filled with 7 μg/ml sulforhodamine B in deionized water, and fluorescent signals were imaged by the CMOS camera. To estimate the beam extent and thickness, FWHMs along the longitudinal ($x$) and transverse ($y$) axis were calculated. For resolution evaluation experiments, yellow–green fluorescence beads (FluoSpheres, ThermoFisher) with a 200-nm diameter embedded in 1% agarose gel were used. PSFs were calculated by analyzing the $z$-stack images of the beads using the software PSFj (http://www.knoplab.de/psfj/), which resulted in the FWHM values of the $xy$-plane and $z$-axis. For the laser power determination experiment, yellow–green fluorescence beads (FluoSpheres, ThermoFisher) with a 2-μm diameter embedded in 1% agarose gel were used. Using a maximum-intensity projection view of the $z$-stack images, centroid intensities of the beads were measured using the software PSFj. In all these experiments, a laser (InSight Deepsee) with a 925-nm wavelength was used.

**Fish husbandry.** Medaka fish were maintained in freshwater tanks with a water circulating system (LABREED, IWAKI) at 26–28 °C under the condition of 14 h light and 10 h dark cycle (8:30–22:30 light). Fish were fed artemia larvae and a powdered diet daily. The spawned eggs were collected in the morning, and eggs were incubated at 28 °C in a dish filled with diluted artificial seawater (0.03(w/v) %) containing methylene blue. Dechorionation of eggs was performed using a hatching enzyme before microscopy observations. The transparent medaka strain See-Through II (STII) strain (StrainID: MT112)[45], and the transgenic strains Tg (pKIF5Aa-GFP) (StrainID: TG1156) which expressed EGFP in neuronal tissues[35], and d-rR-Tg(beta-actin-loxP-DsRed2-loxP-GFP) (StrainID: TG861) which expressed a red fluorescent protein in whole body[46], were supplied by NBRP Medaka (https://shigen.nig.ac.jp/medaka/). The transgenic strain FLT4-EGFP, which expressed EGFP in lymphatic endothelial cells, was kindly provided from Dr. Deguchi (National Institute of Advanced Industrial Science and Technology). All experiments were conducted in accordance with the guidelines of the ethics committee for animal experiments of Ehime University.

**Preparation of medaka embryos and larvae.** For DSLM observations, we embedded fish samples into 1 (w/v)% agarose gel set in the homemade sample holder as described above. Living larvae, 2 weeks after hatching, were first anesthetized by cold water, and then embedded in the agarose gel. We removed the gel around the head to gill using fine forceps to allow fish to breath freely. For DSLM observations, the gel-embedded Medaka samples were put into the sample chamber filled with the diluted artificial seawater medium kept at 28 °C. The medium was aerated and perfused to maintain the water temperature using a tube-heater (TPiE-TH, Tokai Hit). For labeling of blood, an appropriate amount of 10 mg/ml dextran-conjugated tetramethyl rhodamine, 70,000 MW (D1819, ThermoFisher Scientific) diluted in a phosphate-buffered saline was injected to cardiac ventricle using a microinjection technique. The microinjection was performed using a pneumatic microinjector (IM-11-2, Narishige) and homemade micro-glass needles under the stereomicroscope (M205 FA, Leica). When creating the micro-glass needles, the glass capillaries (G-1, Narishige) were pulled in an appropriate heating condition using the flaming/brown style micropipette puller (P-1000, Sutter Instrument).

To observe living embryos, we embedded the embryos into the agarose gel in the same way as the larvae. Prior to the observation, embryos were dechorionated using the hatching enzyme. To observe the $Ca^{2+}$ dynamics, we used the genetically encoded fluorescent $Ca^{2+}$ indicator GCaMP6f[47]. The plasmid DNA solution at the concentration of 20 ng/μL was injected into cell cytoplasm at the one-cell stage STII Medaka embryos. The microinjection was performed similar to dye injection to larvae. The plasmid vector pGP-CMV-GCaMP6f was purchased from Addgene (Addgene plasmid #40755).

To observe $Ca^{2+}$-dependent contraction waves, we implemented a gel cage sample mounting method not to inhibit cellular contraction activities (Supplementary Fig. 8). In this method, the 1% agarose gel mounted with the sample holder was used to make a hole, and then an embryo was placed with medium, and the chamber was capped by the gel. Number of samples used: FLT4-EGFP ×3 for whole body imaging of the lymphatic and blood vessels, KIF5Aa-GFP ×5 for whole brain imaging, beta-actin-DsRed2 ×3 for the different beam condition of phototoxicity experiment, and FLT4-EGFP ×2 for time-lapse analysis of embryonic development, STII ×1 for $Ca^{2+}$ imaging.

**Image processing**

*3D image reconstruction.* 3D/4D image visualization was performed based on a volume-rendering method using Imaris8.4.0 software (Bitplane). Snapshots and movies were also created using Imaris software.

*Stitching of multi z-sequence images.* To visualize whole-body images of Medaka larvae, we used image stitching software implemented as a plugin[34] of Fiji software[48]. A tiled 3D scan of whole body of medaka larvae was performed by DSLM, and then individual $z$-sequence images were used to reconstruct the whole image.

*Image analysis of the $Ca^{2+}$ wave.* To analyze the spatiotemporal mode of the $Ca^{2+}$ wave, we employed coordinate transformation techniques. The observed rhythmic contracting movement was generated with intracellular elevation and intercellular propagation of $Ca^{2+}$. The $Ca^{2+}$ signals associated with the rhythmic contraction waves propagated across the out-most layer of the tissue sheet, which covered both the embryo body and yolk cell. One way to efficiently extract information of the signal dynamics was to exploit the geometry of organisms[42]. Utilizing the advantage of this single surface propagating $Ca^{2+}$ signals, we initially transformed the Cartesian coordinate into the polar coordinate system where the centroid of embryo was set to the origin of the coordinate system. Afterward, every voxel was assigned with $r\theta\varphi$ polar coordinate information. Then the maximum intensity projection with respect to radius shell $r \sim r + \Delta r$ ($r = 300$ μm, $\Delta r = 300$ μm) was performed to map the $Ca^{2+}$ activity onto the plane. Finally, the voxels with $\theta\varphi$ were mapped into the elliptic plane by the Mollweide projection method. This line of image processing is illustrated in Supplementary Fig. 9 and was performed using MATLAB software (R2018b, Mathworks).

**Statistical analyses.** Data are presented as the box plots, the lines of the box indicate the first quantile, median, and third quantile. Upper and lower whiskers indicate the minimum and maximum, respectively. The Cross mark represents the average. The asterisk indicates a $p$ value <0.05 with a one-sided $t$-test. Statistical significance is determined using the Excel software (Office2019, Microsoft).

**Reporting summary.** Further information on research design is available in the Nature Research Reporting Summary linked to this article.

## Data availability

The datasets generated during and/or analyzed during the current study are available in the figshare repository https://doi.org/10.6084/m9.figshare.14229224[33]. Source data are provided with this paper.

## Code availability

The computer codes generated during and/or analyzed during the current study are available from the corresponding author on reasonable request. The MATLAB program

package that performs a numerical integration of the lens-axicon triplet formula is provided as a Supplementary Code.

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

## Acknowledgements

We thank Drs. Bi-Chang Chen (Academia Sinica, Taiwan), Kiyoshi Naruse (National Institute of Basic Biology, Japan), Tomonori Deguchi (National Institute of Advanced Industrial Science and Technology, Japan), and Ryosuke Kawakami (Ehime University, Japan) for helpful discussions. We are grateful to National BioResource Project (NBRP) Medaka for providing the Medaka strains, STII(b) (StrainID: MT112), Tg(pKIF5Aa-GFP) (StrainID: TG1156), d-rR-Tg(beta-actin-loxP-DsRed2-loxP-GFP) (StrainID: TG861) and the hatching enzyme. This work was supported by MEXT/JSPS Grant Number JP19K12218 and JP20H05038 "Cellular Diversity" to T.S., MEXT/JSPS Grant Number JP15H05952 "Resonance Bio" to T.S., and T.I, and MEXT/JSPS KAKENHI Grant Number JP16H06280 "Advanced Bioimaging Support", AMED Grant Number JP20gm1210001 to T.I. This work was also funded by Nakatani Foundation for advancement of measuring technologies in biomedical engineering, and Kato Memorial Bioscience Foundation to T.S.

## Author contributions

S.T. constructed the optical system. T.S. performed the numerical simulations. S.T and T. S performed the experiments. S.T. and T.S. performed data analyses. S.T., T.S, and T.I. wrote the manuscript.

## Competing interests

The authors declare no competing interests.
