## [Peer Review File · Nature Communications]

Reviewers' Comments:

Reviewer #1:

Remarks to the Author:

The present manuscript from Takamezawa, Saitou and Imamura describes an improved method - Two Photon Bessel Beam Illumination - to image biological samples over a long period of time, improving resolution and decreasing toxicity when compared to available methods. The authors revise in a concise but clear manner the main advantages and draw-backs of existing imaging methods, allowing the general reader to get the main novelty that is presented in the manuscript: improving how to image large samples with high temporal and spatial accuracy. The tool is used to image different developmental stages of medaka fish, a model that has previously shown specific challenges related to their slow developmental time and its size at early larval stages. The long term, live imaging that the authors perform on the medaka embryo is remarkable in terms of the resolution of the images, survival of the samples, and neutralisation of the endogenous movements of the samples during the imaging time. Overall, the authors present convincing data using a device that will be useful not just for the medaka community but to other communities working with long term, in vivo imaging.

I have main and minor comments that I would like to be addressed before reaching a final decision on the manuscript.

Main Comments:

In the section "Application to whole body imaging in the medaka larvae", the authors present data (Figure 4) of blood vessels in juveniles. Since the morphology of the system has been previously reported in available atlas (working with fixed sample, though) the section would benefit from including the proper references (Isogai S., Fujita M. (2011) Anatomical Atlas of Blood Vascular System of Medaka. In: Naruse K., Tanaka M., Takeda H. (eds) Medaka. Springer, Tokyo, among others), and from a brief description of what it is known and what it is new here. Any difference that the authors find when comparing in vivo data with previous, fixed data, should be pointed out to illustrate additional aspects in which the approach presented here helps representing a system minimising artefacts introduced when handling biological samples. Also on this point, it would be helpful for the reader to have access to the 3D data acquired by the authors, in the form of an additional Movie perhaps, or the raw data deposited in a web repository.

Data presented at the beginning of the "Long-term observation of embryonic development" section (lines 169 - 177, page 7) is of paramount importance to anyone interested in long term, in-vivo imaging. It would make most sense to have these presented as a main Figure (rather than the actual Sup.Fig 6).

Two main comments on this point are: 1) the panels of Figure 6 look far from optimal. Since the authors report on an imaging method, it would be make most sense that they present data were the embryos, or the structures they want to point out to, are clear to the reader. Including schemes might help here as well. 2) The authors report 180 minutes as the time at which embryos imaged with a Gaussian beam irradiation die. Doing long term in vivo imaging with light sheath technology, however, is possible well beyond that lapse. The authors should be more careful here and do a comparison between Gaussian and Bessel methods using a series of different parameters for which Gaussian methods can be used to image longer times. Otherwise, this feels as an understatement of current imaging methods to give an extra value to their own.

The number of experiments for the photo-toxicity assay seem to be very low (N=2). Is that 2 embryos for each, Gaussian and Bessel, methods? Can this be explained better in M&Ms, and also increase the N for this particular point?

In Methods, the authors state the embryos were dechorionated "in some cases" using hatchling enzyme. It is not clear what they mean with that. Were embryos dechorionated in every case, some times with HE and some times with other methods? Or were they imaged through the chorion? This is an extremely important technical detail that should be properly defined. If embryos can be imaged through the chorion, this would be a very big plus for the technique. Also, there is no reference to any drug added to the medium in order to avoid muscular movements of

the embryos. How was that achieved? Was that problem solved during post-acquisition editing? The authors should be more explicit about it either on the results or in M&M section.

In the last part of the results section, the authors show that they can track the calcium waves typically occurring in the early medaka embryo. This is typically a main draw-back to image embryos at this stage, and the authors use it to follow Ca waves. The description in the main text on data presented in Figure 6 is, however, too shallow, and the authors do not elaborate on this, although they have indeed acquired the data to do so, which seems the most difficult and challenging part of the process. I would encourage them to analyse the data they have acquire and present the conclusion of their imaging.

Minor Comments:

m1) The authors do a good job in the second paragraph of the introduction, where they state the pros and cons of confocal microscopy and light-sheet based techniques. The paragraph would benefit from the inclusion of references within lines 36 and 49, page 3, to complement the ones the authors refer to towards the end of the paragraph.

m2) In the introduction, the authors do not comment on their choice to image medaka. The chosen animal model comes out of the blue in line 72, page 4. The general reader will benefit from a brief description of the medaka embryo, stressing its size (ca. 1mm for the chorion diameter) and the length of its embryonic development (ca. 3x the time of zebrafish) as challenges for the in-vivo imaging approaches. The current description focuses on rather positive aspects and underestimates the challenges of 4D imaging using medaka.

m3) Fig4A - it would help if the scale bar states mm rather than μm .

Reviewer #2:

Remarks to the Author:

The authors present a tunable illumination module for a light sheet microscope capable of producing Bessel beams of various lengths for two-photon excitation. Two-photon Bessel beams have been reported previously by several labs owing to their combination of a long pseudo non-divergent propagation length and minimal contrast loss via the Bessel beam rings via the non-linearity of the excitation process. The unique aspect of this technical implementation is the tuneability of the Bessel beam without the use of spatial light modulators, which are lossy with respect to laser power throughput (this can be problematic owing to the high power requirements of two-photon excitation). To my knowledge this is the first time such a system has been reported. I expect the system to be reproducible by others if desired.

Nevertheless, while the system constitutes a worthy addition to the range of reported two-photon and Bessel beam light sheet microscopes, neither the technical demonstration nor the application seem sufficiently substantial to warrant publication in Nature Communications in their current form. Ultimately, I do not envisage the reported system to majorly influence thinking in the field of light sheet microscopy, owing to the fact that the major aspects have been reported previously several times (Two-photon Bessel excitation) e.g. Planchon et al., Nature Methods 2011, Fahrbach et al., Optics Express, 2013, Welf et al., Dev. Cell 2016

In particular the comparison between different beam modes is flawed and the need for the technology in the context of the application is unclear. For example, there is still a significant worsening of image quality on the distal side of the embryo (relative to the light sheet). Why employ this approach over e.g. two-sided illumination and image fusion, which will provide superior coverage of the far side of the embryo and is technical simple to implement and commercially available otherwise. There may be some benefit of the reported approach for imaging in the center of larger and substantially less transparent specimens that are not amenable to multi-view imaging but this is not reported.

I would urge the authors to consider these points to improve the paper and which may justify a

recommendation for acceptance in the future:

Line 55: The authors mention that the Bessel beam is needle-like without making mention of the surrounding rings. This is actually a justification for two-photon excitation since it suppresses contrast degradation from the rings that is the biggest issue associated with Bessel beam light sheet microscopy. It is also important when considering photo damage and the basic argument for the self-reconstruction after occlusions. The authors should describe the Bessel beam a little more and some of its benefits/drawbacks.

Line 60/70: Any mention of beam FOV is meaningless without knowing the thickness of the main lobe and the energy distribution in the core vs. rings. One can make a Gaussian beam to cover an arbitrary distance with minimal divergence but it will be thick.

Line 61: The aim should be framed more in terms of the needs of the biological imaging to be performed. The link between the biology and optics seems tenuous throughout.

Line 134: The authors note that the intensity of the Gaussian beam at focus is 42x higher than for the Bessel beam. This is of course to be expected. The authors are comparing Gaussian beams with a much shorter focus than the Bessel beam. This is not a fundamental issue of Gaussian vs. Bessel but rather reflects the beam shaping in the two cases. The Bessel beam should be compared against a Gaussian beam of equivalent length to assess the performance for light sheet microscopy. This may help highlight the improvement in axial resolution for the Bessel beam. I refer again to the point made with regard to lines 60/70.

Line 159: Why are thinner shadow lines preferable? For both Gaussian and Bessel beams there are substantial stripe artifacts that could be remedied in a variety of other ways e.g. mSPIM/mDSLM (Huisken et al./Glaser et al. respectively) amongst many others.

Line 168: This section again suffers from the Gaussian beam being generated with a much higher peak intensity at the focus rather than setting the two beam modes on a level playing field (the authors should include laser power at the sample or back focal plane of the illumination objective. If the two beam lengths are matched the Gaussian should outperform the Bessel beam for a given rate of fluorescence signal generation, since the core of the Bessel beam, which generates the majority of the measured signal (owing to non-linearity) contains only a small fraction of the total beam power. i.e. the regions exposed to light from the outer lobes experience elevated light intensities without contributing to signal. This is the tradeoff with Bessel beams.

Line 249: The authors note tuning the longitudinal extent from 600 - 1000 μm . The FWHM used is actually a poor indicator of useable light sheet length. Typically an intensity variation of $\pm 10\%$ across the sheet is acceptable. Nevertheless, the authors provide little to no justification for this tuneability. Multiple different applications within Medaka fish or other samples utilizing different light sheet lengths would go some way to doing so.

Letter to Reviewer#1

The present manuscript from Takamezawa, Saitou and Imamura describes an improved method - Two Photon Bessel Beam Illumination - to image biological samples over a long period of time, improving resolution and decreasing toxicity when compared to available methods. The authors revise in a concise but clear manner the main advantages and draw-backs of existing imaging methods, allowing the general reader to get the main novelty that is presented in the manuscript: improving how to image large samples with high temporal and spatial accuracy. The tool is used to image different developmental stages of medaka fish, a model that has previously shown specific challenges related to their slow developmental time and its size at early larval stages. The long term, live imaging that the authors perform on the medaka embryo is remarkable in terms of the resolution of the images, survival of the samples, and neutralisation of the endogenous movements of the samples during the imaging time.

Overall, the authors present convincing data using a device that will be useful not just for the medaka community but to other communities working with long term, in vivo imaging.

I have main and minor comments that I would like to be addressed before reaching a final decision on the manuscript.

We would like to express our sincerest gratitude for your valuable comments. Based on your comments, we conducted additional experiments regarding Gaussian beams and phototoxicity. In addition, we substantially changed the text and figures. We also added a general description of medaka fish and compared the blood vessel anatomical features between the previous report and our results. We believe that these changes enhanced the description from a practical imaging application viewpoint. Red text in the revised manuscript denotes a modification.

Main Comments:

In the section “Application to whole body imaging in the medaka larvae”, the authors present data (Figure 4) of blood vessels in juveniles. Since the morphology of the system has been previously reported in available atlas (working with fixed sample, though) the section would benefit from including the proper references (Isogai S., Fujita M. (2011) Anatomical Atlas of Blood VascularSystem of Medaka. In: Naruse K., Tanaka M., Takeda H. (eds) Medaka. Springer, Tokyo, among others), and from a brief description of what it is known and what it is new here. Any difference that the authors find when comparing in vivo data with previous, fixed data, should be pointed out to illustrate additional aspects in which the approach presented here helps representing

a system minimising artefacts introduced when handling biological samples.

Also on this point, it would be helpful for the reader to have access to the 3D data acquired by the authors, in the form of an additional Movie perhaps, or the raw data deposited in a web repository.

We added additional descriptions about the anatomical features of the blood vessel of medaka in the Results and Discussion. Not only do we describe the structures of the blood vessel in more detail by focusing on specific tissues, body trunk, tail, and brain, we also compare with the previously reported results and cite additional references, including the one that you suggested. Please see the “Application to whole body imaging of medaka larvae” section (line 216, page 8), Discussion (line 365, page 13), and Supplementary Fig 7. Although the previous analyses were on adult fish and embryos while our analyses are on larvae, there are some basic structures in common. However, there is a difference in the vascular plexus of the spinal cord located dorsal of the vertebrae (Isogai and Fujita, 2011). Live samples should clarify whether fixed samples have significant artifacts. This is an interesting point and will be the subject of future research.

We agree that adding 3D image data to a repository is important. Hence, we are planning to deposit our raw image data shown in Figs. 4A, 4B, 6, and 7B to a community-recognized repository.

Data presented at the beginning of the “Long-term observation of embryonic development” section (lines 169 - 177, page 7) is of paramount importance to anyone interested in long term, in-vivo imaging. It would make most sense to have these presented as a main Figure (rather than the actual Sup.Fig 6).

Two main comments on this point are: 1) the panels of Figure 6 look far from optimal. Since the authors report on an imaging method, it would be make most sense that they present data were the embryos, or the structures they want to point out to, are clear to the reader. Including schemes might help here as well. 2) The authors report 180 minutes as the time at which embryos imaged with a Gaussian beam irradiation die. Doing long term in vivo imaging with light sheath technology, however, is possible well beyond that lapse. The authors should be more careful here and do a comparison between Gaussian and Bessel methods using a series of different parameters for which Gaussian methods can be used to image longer times. Otherwise, this feels as an understatement of current imaging methods to give an extra value to their own.

The number of experiments for the photo-toxicity assay seem to be very low (N=2). Is that 2 embryos for each, Gaussian and Bessel, methods? Can this be explained better in M&Ms, and also increase the N for this particular point?

We appreciate this comment. This is an important point for in vivo imaging applications. For comparison of different beam conditions, we first setup Gaussian beams with different propagation lengths and thicknesses. These are highlighted in the “Comparison of Gaussian and Bessel pathways”

section (line 187, page 7), and Fig. 3, Supplementary Fig. 5 and 6. The Gaussian beams in this revised manuscript have an equivalent length to the Bessel beam. Using these, we performed a new analysis on the phototoxicity assessment. To systematically compare the photodamage effect for different conditions of Gaussian and Bessel beams, we assessed the survival of embryos by 3D time-lapse imaging (line 244, page 9 and Table 1). The imaging results are presented in Fig. 5, which make the structures more recognizable. The number of embryos used for each experiment is 3, as indicated in Table 1.

In Methods, the authors state the embryos were dechorionated “in some cases” using hatching enzyme. It is not clear what they mean with that. Were embryos dechorionated in every case, some times with HE and some times with other methods? Or were they imaged through the chorion? This is an extremely important technical detail that should be properly defined. If embryos can be imaged through the chorion, this would be a very big plus for the technique. Also, there is no reference to any drug added to the medium in order to avoid muscular movements of the embryos. How was that achieved? Was that problem solved during post-acquisition editing? The authors should be more explicit about it either on the results or in M&M section.

In the original version of the manuscript (Supplementary Fig 6), we performed imaging through the chorion, but the detected fluorescence signal was blurred, making it difficult to recognize specific structures. Therefore, we conducted a new experiment with dechorionated embryos in which the ablated structures can be seen clearly (Fig. 5). Hence, we modified the expression regarding dechoriation using hatching enzyme (line 545, page 19).

We did not use any drugs such as tricaine to suppress muscular movements. The embryos and larvae, except for the experiments about embryonic rhythmic contraction movements in Fig. 7, were embedded in 1% agarose gel, which is sufficient to suppress movements during imaging experiments.

In the last part of the results section, the authors show that they can track the calcium waves typically occurring in the early medaka embryo. This is typically a main draw-back to image embryos at this stage, and the authors use it to follow Ca waves. The description in the main text on data presented in Figure 6 is, however, too shallow, and the authors do not elaborate on this, although they have indeed acquired the data to do so, which seems the most difficult and challenging part of the process. I would encourage them to analyse the data they have acquire and present the conclusion of their imaging.

We added a detailed description of this rhythmic contraction movement (line 305, page 11). The purpose of this section is to demonstrate that fast 4D imaging is possible. We chose to track a rapidly propagating extraembryonic calcium wave. We successfully captured this fast-dynamic event with modified recording modes. This description is added to the last sentence of this subsection (line 316,

page 11).

Minor Comments:

m1) The authors do a good job in the second paragraph of the introduction, where they state the pros and cons of confocal microscopy and light-sheet based techniques. The paragraph would benefit from the inclusion of references within lines 36 and 49, page 3, to complement the ones the authors refer to towards the end of the paragraph.

We added references about confocal microscopy (Pawley 2006) (line 38, page 3) and multi-photon microscopy (Denk 1990; Zipfel 2003) (line 80, page 4).

m2) In the introduction, the authors do not comment on their choice to image medaka. The chosen animal model comes out of the blue in line 72, page 4. The general reader will benefit from a brief description of the medaka embryo, stressing its size (ca. 1mm for the chorion diameter) and the length of its embryonic development (ca. 3x the time of zebrafish) as challenges for the in-vivo imaging approaches. The current description focuses on rather positive aspects and underestimates the challenges of 4D imaging using medaka.

We added a general description of medaka fish, including the challenges of using for in vivo imaging.

m3) Fig4A - it would help if the scale bar states mm rather than μm .

This point is corrected in Fig. 4A.

Below is a summary of the corrections in the figures:

Fig. 2 G and H modified

Fig. 3 modified

Fig. 4 C D E F modified

Fig. 5 added newly

Fig. 6 changed the number, but it is the same as original Fig. 5

Fig. 7 changed the number, but it is the same as original Fig. 6

Supplementary Fig. 4–8 added newly

Supplementary Fig. 9, 10 changed the number, but they are the same as original Supplementary Fig. 7, 8
Supplementary Fig. 4–6 deleted

Minor corrections:

Fig. 2, Supplementary Fig.2 and 3, labeling of x and y axis, d1 and d2, is corrected

Fig. 2F, the unit μm is corrected to mm.

Bar plots with sd are converted to box plots as per the journal's policy.

Mistakes in the calculation on Fig. 2H, which did not affect the conclusion, are corrected.

Letter to Reviewer#2

The authors present a tunable illumination module for a light sheet microscope capable of producing Bessel beams of various lengths for two-photon excitation. Two-photon Bessel beams have been reported previously by several labs owing to their combination of a long pseudo non-divergent propagation length and minimal contrast loss via the Bessel beam rings via the non-linearity of the excitation process. The unique aspect of this technical implementation is the tunability of the Bessel beam without the use of spatial light modulators, which are lossy with respect to laser power throughput (this can be problematic owing to the high power requirements of two-photon excitation). To my knowledge this is the first time such a system has been reported. I expect the system to be reproducible by others if desired.

Nevertheless, while the system constitutes a worthy addition to the range of reported two-photon and Bessel beam light sheet microscopes, neither the technical demonstration nor the application seem sufficiently substantial to warrant publication in Nature Communications in their current form. Ultimately, I do not envisage the reported system to majorly influence thinking in the field of light sheet microscopy, owing to the fact that the major aspects have been reported previously several times (Two-photon Bessel excitation) e.g. Planchon et al., Nature Methods 2011, Fahrbach et al., Optics Express, 2013, Welf et al., Dev. Cell 2016

In particular the comparison between different beam modes is flawed and the need for the technology in the context of the application is unclear. For example, there is still a significant worsening of image quality on the distal side of the embryo (relative to the light sheet). Why employ this approach over e.g. two-sided illumination and image fusion, which will provide superior coverage of the far side of the embryo and is technical simple to implement and commercially available otherwise. There may be some benefit of the reported approach for imaging in the center of larger and substantially less transparent specimens that are not amenable to multi-view imaging but this is not reported.

We would like to express our sincere appreciation for your thoughtful and constructive comments. As per your comments, we substantially revised the Introduction and conducted additional experimental analyses of the Gaussian beams and phototoxicity. We also evaluated the tunability and self-reconstruction effect of the Bessel beam. Accordingly, we revised the text and figures. We believe your comments have enhanced the logic and importance of this work. The modified sentences are highlighted as red-colored text in the manuscript.

I would urge the authors to consider these points to improve the paper and which may justify a recommendation for acceptance in the future:

Line 55: The authors mention that the Bessel beam is needle-like without making mention of the surrounding rings. This is actually a justification for two-photon excitation since it suppresses contrast degradation from the rings that is the biggest issue associated with Bessel beam light sheet microscopy. It is also important when considering photo damage and the basic argument for the self-reconstruction after occlusions. The authors should describe the Bessel beam a little more and some of its benefits/drawbacks.

We appreciate this comment. This is a key point for the use of a two-photon Bessel beam that we omitted. We modified the text to provide a more general introduction of the Bessel beam, including its shape, trade-off, and self-reconstruction property. We also added a description of how two-photon excitation improves image contrast degradation due to the side lobes of the Bessel beam (line 76, page 4).

Line 60/70: Any mention of beam FOV is meaningless without knowing the thickness of the main lobe and the energy distribution in the core vs. rings. One can make a Gaussian beam to cover an arbitrary distance with minimal divergence but it will be thick.

We agree with this comment. It is meaningless to mention FOV only because there is a trade-off between the length and transverse distribution of the beam. To address this, we rewrote the Introduction to explain the core and surrounding energy distributions as well as their trade-off. Please see line 50 (page 3) for a general (Gaussian) description and line 69 (page 4) for description of the Bessel beam. As you aptly noted, an advantage of using two-photon excitation is the suppression of out-of-focus signals generated by the side lobes of the Bessel beam. After describing these, we added sentences to highlight the importance of our system for large field of view illumination (line 87, page 4).

Line 61: The aim should be framed more in terms of the needs of the biological imaging to be performed. The link between the biology and optics seems tenuous throughout.

We modified the text to include a general description of medaka fish and described why a wide field imaging technique is necessary to analyze this organism (line 103, page 5).

Line 134: The authors note that the intensity of the Gaussian beam at focus is 42x higher than for the Bessel beam. This is of course to be expected. The authors are comparing Gaussian beams with a much shorter focus than the Bessel beam. This is not a fundamental issue of Gaussian vs. Bessel but

rather reflects the beam shaping in the two cases. The Bessel beam should be compared against a Gaussian beam of equivalent length to assess the performance for light sheet microscopy. This may help highlight the improvement in axial resolution for the Bessel beam. I refer again to the point made with regard to lines 60/70.

To compare various beam conditions, we conducted additional experiments on the Gaussian beams. We setup three types of Gaussian beams with different propagation lengths and thicknesses. We included a Gaussian beam with an equivalent length to the Bessel beam. We then compared the basic beam profiles between these and Bessel beam. To reflect this point in the text, we modified the section “Comparison of Gaussian and Bessel pathways”, (line 187, page 7), and Fig. 3, Supplementary Fig. 5 and 6. In addition, we eliminated Supplementary Figs. 4 and 5 in the original manuscript, which depicted the intensities of Gaussian and Bessel beams, because they did not contain useful information.

Line 159: Why are thinner shadow lines preferable? For both Gaussian and Bessel beams there are substantial stripe artifacts that could be remedied in a variety of other ways e.g. mSPIM/mDSLIM (Huisken et al./Glaser et al. respectively) amongst many others.

Stripe artifacts can be rescued by mSPIM, multi-view imaging, or structured illumination. Their descriptions are added to the Introduction (line 59, page 3). An alternative approach to improve the image contrast is to use the self-reconstruction property associated with the Bessel beam. A Gaussian beam generates illumination artifacts in the form of stripes, while the Bessel beam reduces this artifact and achieves a more homogeneous image with lower fluctuations and weakened stripe patterns. To evaluate this image homogeneity, we previously analyzed the power ratio between the high and low frequencies (Fahrbach 2013). To see this from another viewpoint, in the revised manuscript, we also analyzed the flatness of the line profiles (i.e., standard deviation) similar to Fahrbach et al. (2010). These analyses might provide firm evidence for image contrast improvement. To reflect these points, we modified the manuscript (line 229, page 9) and Figs. 4D-F.

Line 168: This section again suffers from the Gaussian beam being generated with a much higher peak intensity at the focus rather than setting the two beam modes on a level playing field (the authors should include laser power at the sample or back focal plane of the illumination objective. If the two beam lengths are matched the Gaussian should outperform the Bessel beam for a given rate of fluorescence signal generation, since the core of the Bessel beam, which generates the majority of the measured signal (owing to non-linearity) contains only a small fraction of the total beam power. i.e. the regions exposed to light from the outer lobes experience elevated light intensities without contributing to signal. This is the tradeoff with Bessel beams.

This section is an important to in vivo imaging applications. As commented, our previous experiments did not employ conditions suitable to evaluate the phototoxicity in a Gaussian or Bessel beam. Here, we performed a new experiment with Bessel and Gaussian beams, which we set up as noted above. To compare the photodamage effect between the Gaussian and Bessel beams in a more systematic way, we assessed the survival of embryos by 3D time-lapse imaging (line 244, page 9 and Table 1). The imaging results are presented in Fig. 5. In this experiment, the laser powers of Bessel and Gaussian beams at the back focal plane of IO were measured and set to 500 mW or 300 mW (line 434, page 15).

Line 249: The authors note tuning the longitudinal extent from 600 - 1000 μm . The FWHM used is actually a poor indicator of useable light sheet length. Typically an intensity variation of $\pm 10\%$ across the sheet is acceptable. Nevertheless, the authors provide little to no justification for this tuneability. Multiple different applications within Medaka fish or other samples utilizing different light sheet lengths would go some way to doing so.

As a confirmation of the tunability of the Bessel beam, we calculated the excitation confinement proposed in Welf (2016), which indicates how much the excitation signal is concentrated within the depth of focus of DO. Thus, a decrease in this value indicates an increase in the out-of-focus blur, which reduces the image quality. In contrast, consistency of this value means a stable optical sectioning capability. We consider this to be sufficient to evaluate the tunability for the Bessel beam length. The results confirm that this ratio is constant for various Bessel beams, while that of the Gaussian beam changes with the length (line 166, page 7, Fig. 2G, line 194, page 8, and Fig. 3C).

The FWHM has been used in several works, including (Welf 2016, Olarte 2012, and Zhao 2014) to express the propagation length of Bessel beams for a two-photon excitation. Therefore, we used it as an indicator of the beam length. We use the FOV throughout the manuscript. To realize a consistent description of the FOV (we previously used the camera window as FOV, which was confusing), we modified the text in line 23 (page 2) and line 341 (page 12). In addition, we eliminated the sentence “Using the 10x magnification...” (line 357, page 14) in the original version of the manuscript for a consistent description.

Whole brain imaging for different Bessel and Gaussian beams is provided in Supplementary Fig. 8.

Below is a summary of the corrections in the figures:

Fig. 2 G and H modified

Fig. 3 modified

Fig. 4 C–F modified

Fig. 5 added newly

Fig. 6 changed the number, but it is the same as original Fig. 5

Fig. 7 changed the number, but it is the same as original Fig. 6

Supplementary Fig. 4–8 added newly

Supplementary Fig. 9, 10 changed the number, but they are the same as original Supplementary Fig. 7, 8

Supplementary Fig. 4–6 deleted

Minor corrections:

Fig. 2, Supplementary Fig. 2 and 3, labeling of x and y axis, d1 and d2, is corrected

Fig. 2F, the unit μm is corrected to mm.

Bar plots with sd are converted to box plots in accordance with the journal's policy.

Mistakes in the calculation on Fig. 2H, which did not affect the conclusion, are corrected.

Reviewers' Comments:

Reviewer #1:

Remarks to the Author:

The authors have addressed most of the issues raised by myself on their first version of the manuscript. I am now please with the new statements and measurement added to the initial submission.

I have two points to mention:

There is one phrase that still seems an overstatement, considering the data that is presented. This is in the Discussion, line 329, pg 13 "This allows whole mount body imaging of a Medaka fish at a single cell resolution". The transgenic lines used by the authors label entire systems or big groups of cells with non-localised fluorescent proteins. The manuscript does not contain any data in which single cells are (or could be) quantified in a given set up. I would encourage the authors to tone down the mentioned statement, on the lines of "This allows whole mount body imaging of a Medaka juvenile with cellular resolution"

 I still consider that adding the 3D data to the repository is important, and I would urge the journal to link it to the original article.

Minor point: the convention indicates that fish should be shown with their anterior side to the left. Please use consistent standards in all panels of figure 4.

Reviewer #2:

Remarks to the Author:

The additional work on behalf of the authors has generally improved the manuscript and addressed some of the concerns around the claims made. However, the assessment of phototoxicity remains flawed without providing additional data concerning image quality. Since a reduction in phototoxicity (relative to two-photon Gaussian excitation at least) is a major claim, the current data presented are simply insufficient to warrant publication at this time.

The authors compare the two Gaussian beams and the Bessel beams at 300 mW and 500 mW for single and multiple acquisitions. However, these beam modes must be compared in the context of their ability to generate useful contrast. For example, if the central lobe of the Bessel beam carries just a quarter of the power (across any cross-section), then the excitation efficiency will be just 1/16 of the case for a Gaussian with equivalent main lobe width (due to the quadratically dependent excitation efficiency for two-photon processes). As such, the Bessel mode would be expected to require substantially higher laser powers to yield an equivalent number of photons emitted. Even when factoring in the fact that the Gaussian C of roughly equivalent usable length to the Bessel mode has slightly lower excitation confinement (hence slightly fewer of the collected photons contribute positively to the image formation) the difference in useful photons per unit laser power would still be expected to be much lower in the Bessel case.

I suggest that the authors determine an optimum laser power for the Bessel and Gaussian C modes that generate images of equivalent signal to noise or contrast. Otherwise comparing a Gaussian mode at 300 mW and a Bessel mode at 300 mW is essentially meaningless.

Beyond this, my major concern with the manuscript, which was noted in the original review was the lack of example applications illustrating that the tuneable Bessel length using the scheme could be useful for imaging different regions within tissue, while showing that this is beneficial to optimize image quality/reduce phototoxicity etc. This, alongside the unique way of generating tuneable Bessel beams, would provide a meaningful development beyond similar work e.g. Meinert & Rohrbach, Biomedical Optics Express, 2019. This aspect was not addressed

Letter to Reviewer#1

The authors have addressed most of the issues raised by myself on their first version of the manuscript. I am now please with the new statements and measurement added to the initial submission.

We would like to express our sincere appreciation for your comments. According to the comments, we corrected the text.

I have two points to mention:

There is one phrase that still seems an overstatement, considering the data that is presented. This is in the Discussion, line 329, pg 13 “This allows whole mount body imaging of a Medaka fish at a single cell resolution”. The transgenic lines used by the authors label entire systems or big groups of cells with non-localised fluorescent proteins. The manuscript does not contain any data in which single cells are (or could be) quantified in a given set up. I would encourage the authors to tone down the mentioned statement, on the lines of “This allows whole mount body imaging of a Medaka juvenile with cellular resolution”

As per your suggestion, this point is corrected. In addition, we modified the sentence on line 120 (page 5) for a consistent description.

I still consider that adding the 3D data to the repository is important, and I would urge the journal to link it to the original article.

We will deposit our raw image data to a community-recognized repository in the very near future.

Minor point: the convention indicates that fish should be shown with their anterior side to the left. Please use consistent standards in all panels of figure 4.

This is revised to conform with convention.

Letter to Reviewer#2

The additional work on behalf of the authors has generally improved the manuscript and addressed some of the concerns around the claims made. However, the assessment of phototoxicity remains flawed without providing additional data concerning image quality. Since a reduction in phototoxicity (relative to two-photon Gaussian excitation at least) is a major claim, the current data presented are simply insufficient to warrant publication at this time.

The authors compare the two Gaussian beams and the Bessel beams at 300 mW and 500 mW for single and multiple acquisitions. However, these beam modes must be compared in the context of their ability to generate useful contrast. For example, if the central lobe of the Bessel beam carries just a quarter of the power (across any cross-section), then the excitation efficiency will be just 1/16 of the case for a Gaussian with equivalent main lobe width (due to the quadratically dependent excitation efficiency for two-photon processes). As such, the Bessel mode would be expected to require substantially higher laser powers to yield an equivalent number of photons emitted. Even when factoring in the fact that the Gaussian C of roughly equivalent usable length to the Bessel mode has slightly lower excitation confinement (hence slightly fewer of the collected photons contribute positively to the image formation) the difference in useful photons per unit laser power would still be expected to be much lower in the Bessel case.

I suggest that the authors determine an optimum laser power for the Bessel and Gaussian C modes that generate images of equivalent signal to noise or contrast. Otherwise comparing a Gaussian mode at 300 mW and a Bessel mode at 300 mW is essentially meaningless.

We would like to express our sincere appreciation for your comments. As per your comments, we conducted additional experimental analysis of the phototoxicity. As suggested, we determined the average laser power for the Gaussian C optics as 280 mW, which generates an equal fluorescent peak intensity as Bessel beams of 500 mW (Supplementary Fig. 10). We then conducted a survival assessment using these laser settings. No differences in the embryo survival rate were observed when the embryos were subjected to 20-times z-sequence measurements for 280 mW Gaussian C beam and 500 mW Bessel beam irradiation. Thus, we proceeded with the assessment. We sought to identify conditions that affected survival. We tested the 565 mW Bessel beam, and found 20-times time-lapse measurements influenced the survival of embryos, but 10-times did not. We then tested the corresponding Gaussian beam (323 mW), which resulted in significant damage in the 10-times measurement. This difference indicates that the peak intensity and the beam thickness are both critical to the survival since the thickness of the Gaussian is broader than that of the Bessel beam when equalizing the longitudinal beam extent. Therefore, these results might provide a firm evidence for the lower phototoxicity of Bessel beam. These results are described from line 274 (page

11) and Table 1, and a discussion is added in line 398 (page 15). The modified sentences are highlighted as red-colored text in the manuscript.

Beyond this, my major concern with the manuscript, which was noted in the original review was the lack of example applications illustrating that the tuneable Bessel length using the scheme could be useful for imaging different regions within tissue, while showing that this is beneficial to optimize image quality/reduce phototoxicity etc. This, alongside the unique way of generating tuneable Bessel beams, would provide a meaningful development beyond similar work e.g. Meinert & Rohrbach, *Biomedical Optics Express*, 2019. This aspect was not addressed

As per this comment, we modified the text and figures based on previous and additional experimental data to confirm the applicability of tunable Bessel beams in bio-imaging. As already shown, our tunable Bessel beams can cover FOV 600–1000 μm with similar spatial resolution values. However, the degrees of self-reconstruction, which contribute to the quality, differ among these. This suggests that the adaptation of the beam extent to the sample length can optimize the image quality. To make this point clear in the manuscript, we added Fig. 4 as a new figure for the evaluations of self-reconstruction, and rewrote the first paragraph of the section “Application to live imaging of Medaka” from line 210 (page 9). We first compared Bessel and Gaussian beams, which showed a significant difference in the quantitative measures of self-reconstruction (same in the previous revision). Then, we added new comparison data between Bessel beams, indicating that the measures are improved in a shorter extent Bessel beam (Figures 4E and 4F). To further show examples, embryo and larva imaging results with different beam setups were added in Supplementary Fig. 7. The tunability of the beam extent helps to acquire a whole shot of embryos to juveniles with improved image quality, and which allow us to perform fast time-lapse imaging over the entire region as demonstrated in capturing Ca^{2+} wave rounding whole outer layer of an embryo (Fig. 8). We consider that these data sufficient representations of application examples. Additionally, to demonstrate advantages for excitation homogeneity and penetration inside tissues, we compared Bessel and Gaussian beams with equivalent beam lengths, and confirmed the image contrast improvement. These are highlighted in Supplementary Fig. 8 and sentences from line 234 (page 10). For comparison of the previous report (Meinert & Rohrbach, *Biomed Opt Exp* 2019), we added sentences from line 390 (page 15). The modified sentences are highlighted as red-colored text in the manuscript.

Reviewers' Comments:

Reviewer #2:

Remarks to the Author:

The authors have made some attempt to address my previous comments including a comparison of the signal generating capability of the two beam modes and a nice demonstration that tuning the beam length can be beneficial for avoiding striped artifacts. While the latter is sufficient, the former is still lacking. Please, see the below comments, which should be addressed in full before publication should be considered.

Line 80: The reported advantages of using a two-photon excitation in light-sheet microscopy are improved penetration depth and background rejection as well as less photodamage [22].

This statement is misleading. The article of Truong et al. claims that two-photon light sheet microscopy leads to reduced photodamage in comparison to point-scanning two-photon microscopy, not single-photon light sheet microscopy. Please remove the claim regarding photo damage, or amend to reflect that the lower peak intensities of light sheet microscopy more generally make it more gentle with respect to photo damage (since photo damage is primarily non-linear in nature).

Line 243: The demonstration of how tuning the beam properties can improve the destriping is sufficient as a proof of principle in the context of this manuscript.

Line 294, Table 1, Suppl. Fig 10. The authors have made some effort to compare photo damage on an even playing field with respect to Gaussian and Bessel illumination. I still believe that the analysis is flawed. Although the 280 mW Gaussian and 500 mW Bessel beam provide the same peak intensity, the integrated intensity will still be substantially reduced in the Bessel case. This is ultimately what matters as the effective dwell time along y is larger for the Gaussian beam. If the authors scan the beam in y and make the same measurements in the rhodamine solution, I would expect that they find that the peak fluorescence intensity would be substantially higher across the FOV for the Gaussian mode (except perhaps at the edges where the effect of the dwell time is reduced due to boundary effects of scanning). First, I would recommend that the authors make these measurements or simply equalise the integrated intensities of the two beam modes. Equalising the power under this condition would provide a fairer comparison than the one provided.

Nevertheless, some of the contribution from the Gaussian beam may in fact arise from out of focus locations, which is hard to determine for a uniformly fluorescent solution. I would also recommend making the measurement on a fluorescent bead suspension to compare useful (i.e. high contrast/in focus) signal against simply the ability to yield signal (regardless of where it is from). This may prove to demonstrate that the required power of the Bessel mode is indeed not so much higher than the Gaussian (the analysis I suggest above would be very likely to show the reverse of this.). This would provide a truly fair comparison of the two modes, following which, the survival studies could be repeated given the relative powers for equivalent useful contrast.

The claims regarding photo damage are very bold and ultimately the rigor here is insufficient to support the authors assertions.

Suppl. Fig 8. The Gaussian mode looks to have superior axial resolution. Is this a result of the tails of the Bessel mode exciting out of plane fluorescence or just a result of the differently scaled brightness of the two images?

The authors have made some attempt to address my previous comments including a comparison of the signal generating capability of the two beam modes and a nice demonstration that tuning the beam length can be beneficial for avoiding striped artifacts. While the latter is sufficient, the former is still lacking. Please, see the below comments, which should be addressed in full before publication should be considered.

Line 80: The reported advantages of using a two-photon excitation in light-sheet microscopy are improved penetration depth and background rejection as well as less photodamage [22].

This statement is misleading. The article of Truong et al. claims that two-photon light sheet microscopy leads to reduced photodamage in comparison to point-scanning two-photon microscopy, not single-photon light sheet microscopy. Please remove the claim regarding photo damage, or amend to reflect that the lower peak intensities of light sheet microscopy more generally make it more gentle with respect to photo damage (since photo damage is primarily non-linear in nature).

We removed the photodamage statement in this sentence as suggested.

Line 243: The demonstration of how tuning the beam properties can improve the destriping is sufficient as a proof of principle in the context of this manuscript.

Thank you very much.

Line 294, Table 1, Suppl. Fig 10. The authors have made some effort to compare photo damage on an even playing field with respect to Gaussian and Bessel illumination. I still believe that the analysis is flawed. Although the 280 mW Gaussian and 500 mW Bessel beam provide the same peak intensity, the integrated intensity will still be substantially reduced in the Bessel case. This is ultimately what matters as the effective dwell time along y is larger for the Gaussian beam. If the authors scan the beam in y and make the same measurements in the rhodamine solution, I would expect that they find that the peak fluorescence intensity would be substantially higher across the FOV for the Gaussian mode (except perhaps at the edges where the effect of the dwell time is reduced due to boundary effects of scanning). First, I would recommend that the authors make these measurements or simply equalise the integrated intensities of the two beam modes. Equalising the power under this condition would provide a fairer comparison than the one provided.

Nevertheless, some of the contribution from the Gaussian beam may in fact arise from out of focus locations, which is hard to determine for a uniformly fluorescent solution. I would also recommend making the measurement on a fluorescent bead suspension to compare useful (i.e. high contrast/in

focus) signal against simply the ability to yield signal (regardless of where it is from). This may prove to demonstrate that the required power of the Bessel mode is indeed not so much higher than the Gaussian (the analysis I suggest above would be very likely to show the reverse of this.). This would provide a truly fair comparison of the two modes, following which, the survival studies could be repeated given the relative powers for equivalent useful contrast.

The claims regarding photo damage are very bold and ultimately the rigor here is insufficient to support the authors assertions.

We appreciate this comment. As per this comment, we conducted additional experiments of the phototoxicity, which might provide more rigorous comparison of the Gaussian and Bessel beams. We previously showed that, using rhodamine solution, 280mW Gaussian and 500mW Bessel beam provide same peak intensity. However, these two modes did not provide same integrated intensity (in case of integrated intensity, 223mW of Gaussian corresponds to Bessel 500mW). In addition, width of Gaussian beam is $\sim 10\mu\text{m}$ which is wider than the depth of focus of DO, thus, as commented, Gaussian beam intensity measured by Rhodamine solution contains out of focus signals. We agreed the reviewer's comment that rigor is insufficient for comparison of the two beam modes. Therefore, as the reviewer suggested, for fair comparison of the in-focus signals between the two beam modes, we used $2\text{-}\mu\text{m}$ diameter green-yellow fluorescent beads, and measured intensities of the beads around the beam peak. We determined that the laser power for the Gaussian C optics of 325 mW generates an equal fluorescent intensity as Bessel beams of 500 mW (Supplementary Fig. 10) (the power of Gaussian estimated here is higher than that estimated previously by Rhodamine solution, as commented). We then conducted a survival assessment using these laser settings. Gaussian beam (325 mW) resulted in significant damage in the 10-times measurement, but Bessel beam (500mW) did not show damage (Table 1). Therefore, these results might provide a firm evidence for our conclusion of the lower phototoxicity of Bessel beam. These results are described from line 274 (page 11) and Table 1. The modified sentences are highlighted as red-colored text in the manuscript.

Suppl. Fig 8. The Gaussian mode looks to have superior axial resolution. Is this a result of the tails of the Bessel mode exciting out of plane fluorescence or just a result of the differently scaled brightness of the two images?

This is a result of a projection of 3D image to xz and yz planes whose widths are defined by the yellow lines. We have previously set wider projection area, which provided a number of overlapped cells in the projection image in Bessel mode. We thus modified this projection area. Images of narrower projection area show separated cells in Bessel mode (Supplementary Fig. 8).

Reviewers' Comments:

Reviewer #2:

Remarks to the Author:

Given the authors most recent and more rigorous comparison of the useful signal generating properties of the two beam modes, I would now recommend the manuscript for publication. Note that the analysis added supports the authors assertions regarding photodamage to a greater degree than the original analysis.